# Internalizing Self-Consistency in Language Models: Multi-Agent Consensus Alignment

## Abstract

Language Models (LMs) are inconsistent reasoners, often generating contradictory responses to identical prompts. While inference-time methods can mitigate these inconsistencies, they fail to address the core problem: LMs struggle to reliably select reasoning pathways that lead to consistent outcomes under exploratory sampling. To address this, we formalize self-consistency as an intrinsic property of well-aligned reasoning models and introduce Multi-Agent Consensus Alignment (MACA), a reinforcement learning framework that post-trains models to favor reasoning trajectories aligned with their internal consensus using majority/minority outcomes from multi-agent debate. These trajectories emerge from deliberative exchanges where agents ground reasoning in peer arguments, not just aggregation of independent attempts, creating richer consensus signals than single-round majority voting. MACA enables agents to teach themselves to be more decisive and concise, and better leverage peer insights in multi-agent settings without external supervision, driving substantial improvements across self-consistency (+27.6% on GSM8K), single-agent reasoning (+23.7% on MATH), sampling-based inference (+22.4% Pass@20 on MATH), and multi-agent ensemble decision-making (+42.7% on MathQA). These findings, coupled with strong generalization to unseen benchmarks (+16.3% on GPQA, +11.6% on CommonsenseQA), demonstrate robust self-alignment that more reliably unlocks latent reasoning potential of language models.

## 1 Introduction

A fundamental trait of a reliable reasoning model is self-consistency: the intrinsic ability to produce stable outputs across various sampled reasoning paths (Elazar et al., 2021; Wang et al., 2022). In the human brain, this consistency emerges from the prefrontal and anterior cingulate cortices, which resolve conflicts between competing neural activations (Miller & Cohen, 2001) by balancing novelty and coherence (Friston, 2010; Botvinick et al., 2004; Shenhav et al., 2013; Zhang et al., 2025). This enables human reasoning to remain robust despite the inherent randomness of thought. In contrast, while probabilistic decoding in language models (LMs) gives access to diverse reasoning trajectories, it struggles to consistently select high-quality paths (Holtzman et al., 2020; Wang et al., 2022). Yet, current AI alignment research primarily focuses on human preferences and external values (Ouyang et al., 2022; Glaese et al., 2022), while overlooking the model's self-alignment. The challenge remains: teaching models to sample diversely, i.e., exploring multiple valid reasoning paths like different theorem proofs or alternative chains of thought, while maintaining consistent quality and conclusions. Existing methods for mitigating sampling inconsistencies such as sampling multiple reasoning paths and aggregating via majority vote (Wang et al., 2022; Li et al., 2024) or using multi-agent debate (Irving et al., 2018) operate at inference time. While these reduce output variance, they do not improve the model's internal reasoning stability. When models generate low-quality reasoning traces, aggregation can even be counterproductive: noisy arguments compound rather than cancel out, especially in ambiguous scenarios (Radharapu et al., 2025).

We formalize self-consistency as an intrinsic property of well-aligned reasoning models and introduce Multi-Agent Consensus Alignment (MACA), a reinforcement learning (RL) framework where multiple LM clones collaborate to solve problems through iterative debate. This debate serves as a simulation environment where agents explore solutions independently, then ground their reasoning through peer interaction and update their answers. Crucially, the reasoning paths exchanged during

this collaborative exploration, not just final majority answers, provide strong training signals for teaching agents to recognize stable reasoning patterns (Fig. 1). These consensus-aligned trajectories from debate contain richer signals than aggregating isolated reasoning attempts. We reinforce majority-outcome traces where agents successfully converged through peer grounding. This teaches models to internalize self-consistency: they learn from collaboratively refined reasoning and develop an inductive bias toward consensus-forming trajectories even mid-generation, which resembles human intuition for sensing sound arguments before completing them (Kahneman, 2011). Learning from these patterns teaches models to weigh multiple reasoning approaches, a skill that generalizes beyond the original training tasks and makes them better ensemble participants.

Our experiments confirm that multi-agent debate produces more informative training signals than single-round majority voting. We also observe that addressing consensus alignment through preference learning yields substantial improvements over scalar-reward RL and imitation learning. We optimize the separation between majority and minority trajectories using majority vote variants of DPO (Rafailov et al., 2023) and KTO (Ethayarajh et al., 2024), outperforming GRPO (Shao et al., 2024) and SFT (Subramaniam et al., 2025). This mirrors human preference formation through relative comparison (Festinger, 1957): when truth is ambiguous, judgments emerge through comparative assessment where majority opinions provide normative pressure while minority views introduce necessary variation (Moscovici, 1976; Nemeth, 1985; M. J. A. N. de Caritat, 1785). Training on debate-derived preferences thus teaches models to ground reasoning in peer arguments, learning efficient and stable reasoning through comparison rather than ground-truth labels.

**Key contributions.** Through extensive experiments on LMs across various reasoning benchmarks, we empirically demonstrate that MACA achieves improvements on the following dimensions.

- **Self-consistency.** MACA shows improvements in answer consistency (up to +27.6% on GSM8K) across different sampled reasoning paths.
- **Accuracy.** It also yields significant improvements in individual agent performance (+23.7% on MATH), sampling-based inference (+22.4% Pass@20 on MATH), and multi-agent performance (+42.7% on MathQA).
- **Generalization.** Training for self-consistency on mathematical reasoning transfers to all evaluated tasks, including unseen domains (+11.3% on GPQA, +11.6% on CommonsenseQA), demonstrating that self-consistency is a foundational capability for general reasoning.

## 2 RELATED WORK

Existing approaches address sampling inconsistency primarily through inference-time techniques. Self-consistency prompting (Wang et al., 2022; Li et al., 2024) samples multiple reasoning paths and selects the majority-voted answer, with extensions for non-verifiable outputs (Chen et al., 2023) and path pruning (Zhu et al., 2024a). Multi-agent debate frameworks (Du et al., 2023; Irving et al., 2018) similarly utilize consensus across models to improve reliability, with recent work exploring applications in scientific discovery (Gottweis et al., 2025). These methods, however, require additional inference compute and do not internalize the self-consistency into the model. We instead improve self-consistency through post-training that optimizes consensus signals via multi-agent RL (Yang et al., 2021; Jiang & Lu, 2021; Zhu et al., 2024b; Zhan et al., 2025), strengthening foundational reasoning abilities. Current training-time alternatives have limitations: relative log-probability ranking (Huang et al., 2025) correlates weakly with accuracy compared to consensus (App. I), while LLM-as-a-Judge approaches (Jiao et al., 2025) suffer from preference leakage (Li et al., 2025) and bias under ambiguity (Radharapu et al., 2025). Majority vote RL methods such as TTRL (Zuo et al., 2025) and ScDPO (Prasad et al., 2024) use GRPO and DPO, respectively, to reinforce single-round majority vote, whereas our framework leverages multi-agent debate, while supporting both preference learning and scalar-reward formulations. Both TTRL and ScDPO are specialized cases of the framework developed in this work, when reducing the multi-agent debate parameters to a single-round majority vote. Subramaniam et al. (2025) uses Supervised Fine-Tuning (SFT) for multi-agent debate optimization. In this work we demonstrate that RL-based alternatives achieve superior performance compared to SFT.

## 3 FORMALIZING SELF-CONSISTENCY

Given a prompt $x$, an LM with parameters $\theta$ defines a distribution $\pi_\theta(y \mid x) = \prod_{t=1}^{|y|} \pi_\theta(y_t \mid x, y_{<t})$ over reasoning trajectories $y$, from which answers $a = A(y)$ are extracted. Under temperature sampling, the model samples from a modified distribution $\pi_{\theta,\tau}(y \mid x)$ where token probabilities are adjusted by temperature $\tau > 0$. This induces an answer distribution $P_{\theta,\tau}(a \mid x) = \sum_{y:A(y)=a} \pi_{\theta,\tau}(y \mid x)$, which gives each answer's probability by summing over all reasoning paths that lead to it. We denote the majority answer as $a^\star_{\theta,\tau}(x) = \arg\max_a P_{\theta,\tau}(a \mid x)$ with majority probability $S^+_{\theta,\tau}(x) = P_{\theta,\tau}(a^\star_{\theta,\tau}(x) \mid x)$. This represents the total probability mass concentrated on the most likely answer, or the model's internal consensus.

Temperature sampling enables exploration of diverse reasoning paths, but reduces the consistency of the final answer. While greedy decoding ($\tau = 0$) trivially approaches perfect consistency, it eliminates exploration and often produces suboptimal solutions (Holtzman et al., 2020). Lower temperatures increase consistency but restrict reasoning diversity. A self-consistent model should maintain high $S^+_{\theta,\tau}(x)$ even at high temperatures, allowing the model to access diverse reasoning trajectories while reliably converging on consistent answers. We measure self-consistency in two ways.

**Single-agent sampling consistency.** Computing $S^+_{\theta,\tau}(x)$ directly requires summing probabilities over all trajectories that lead to the majority answer, which is untractable. Instead, we estimate it by sampling $t$ independent trajectories with answers $a_1, \ldots, a_t$ and computing:

$$s^{\theta,\tau}_t(x) = \frac{1}{t} \sum_{i=1}^{t} \mathbf{1}[a_i(x) = \hat{a}(x)], \quad \text{where } \hat{a}(x) = \text{Majority}\{a_1(x), \ldots, a_t(x)\}. \tag{1}$$

This measures the fraction of sampled trajectories that agree with the majority answer. As $t \to \infty$, $s^{\theta,\tau}_t(x) \to S^+_{\theta,\tau}(x)$, providing a consistent estimate of the true majority probability.

**Multi-agent debate agreement.** When $M$ agents produce answers $a_1(x), \ldots, a_M(x)$ through deliberation, we measure the fraction of agents converging on the majority answer:

$$d^{\theta,\tau}_M(x) = \frac{1}{M} \sum_{m=1}^{M} \mathbf{1}[a_m(x) = \hat{a}(x)], \quad \text{where } \hat{a}(x) = \text{Majority}\{a_1(x), \ldots, a_M(x)\}. \tag{2}$$

Higher agreement indicates a stronger consensus. In other words, models with higher $S^+_{\theta,\tau}(x)$ reach the same conclusion more frequently.

## 4 MACA: MULTI-AGENT CONSENSUS ALIGNMENT

Having formalized self-consistency, we now present a framework to improve it through post-training with self-generated signals from debate. In multi-agent debate, $M$ copies of the same model engage in iterative discussion: each agent generates an initial response, then all agents see each other's reasoning and update their answers over $R - 1$ subsequent rounds of deliberation. Answers that persist indicate stronger reasoning. The framework requires no external supervision: agents supervise themselves by learning from their own debate dynamics. Specifically, for each prompt $x$, the debate produces final responses $\mathcal{Y}(x) = \{y_1, \ldots, y_M\}$ with extracted answers $a_m = \mathcal{A}(y_m)$. The majority consensus $\hat{a}(x) = \text{Majority}\{a_1, \ldots, a_M\}$ partitions $\mathcal{Y}(x)$ into consensus-supporting $\mathcal{G}^+(x) = \{y \in \mathcal{Y}(x) : \mathcal{A}(y) = \hat{a}(x)\}$ and dissenting $\mathcal{G}^-(x) = \{y \in \mathcal{Y}(x) : \mathcal{A}(y) \neq \hat{a}(x)\}$ trajectories. This creates a fixed post-training dataset $\mathcal{D}_{\text{post}} = \{(x, \hat{a}(x), \mathcal{G}^+(x), \mathcal{G}^-(x))\}_{x \in \mathcal{D}}$ where $\mathcal{D}$ is the original set of prompts. Debate consensus, arising through deliberative exchange rather than statistical sampling, provides rich training signals. We adapt four post-training objectives to this self-generated data, treating consensus-supporting trajectories ($\mathcal{G}^+$) as preferred and dissenting trajectories ($\mathcal{G}^-$) as not preferred. By learning to separate these groups, the model internalizes the nuanced differences between stable consensus and dissenting reasoning. See Alg. 1 in App. D for the complete iterative debate and post-training loop.

**Majority-Vote SFT (MV-SFT)** trains the model to mimic consensus-supporting trajectories:

$$\mathcal{L}_{\text{SFT}}(\theta) = -\mathbb{E}_{x \sim \mathcal{D}} \mathbb{E}_{y^+ \in \mathcal{G}^+(x)} [\log \pi_\theta(y^+ \mid x)]. \tag{3}$$

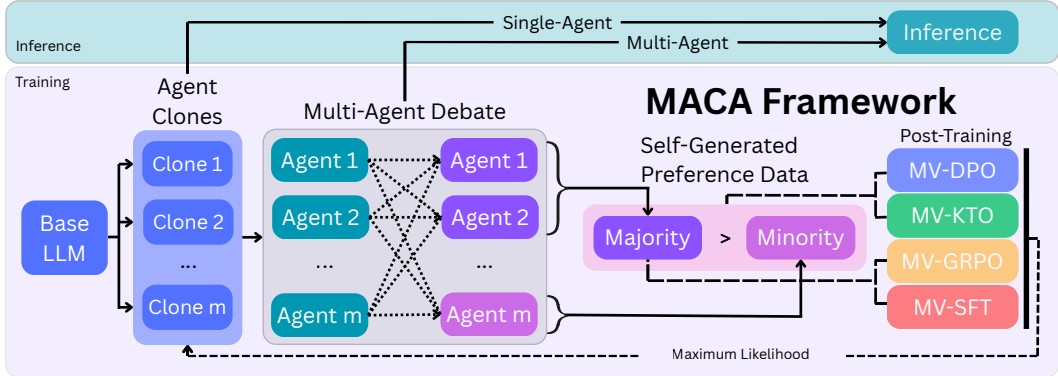

Figure 1: Multi-Agent Consensus Alignment framework: Multiple clones of a base LM engage in debate to generate majority and minority reasoning trajectories through multi-agent debate. The framework splits responses based on alignment with majority consensus to create preference pairs. MV-GRPO compares online samples against majority signals, while MV-SFT imitates majority traces directly. In contrast, MV-DPO and MV-KTO utilize both positive (majority) and negative (minority) examples to learn relative separation between these preference pairs. Updated agents can then be used for single-agent or multi-agent inference, or continue iterative training.

**Majority-Vote GRPO (MV-GRPO)** uses online sampling with consensus-based rewards. For each prompt $x$, we sample multiple trajectories from the current policy and assign reward $r_x(y) = \mathbf{1}[\mathcal{A}(y) = \hat{a}(x)]$ based on whether each sample's answer matches the pre-computed consensus:

$$\mathcal{L}_{\text{GRPO}}(\theta) = -\mathbb{E}_{x \sim \mathcal{D}} \mathbb{E}_{y \sim \pi_\theta} \left[ \tilde{A}_x(y) \sum_t \log \pi_\theta(y_t | x, y_{<t}) \right] + \lambda \, \text{KL}(\pi_\theta \| \pi_{\text{ref}}), \quad (4)$$

where $\tilde{A}_x(y) = r_x(y) - \bar{r}_x$ is the group-normalized advantage. We find that model inconsistency naturally yields both consensus and dissenting trajectories, allowing GRPO's group normalization to contrast majority/minority outcomes within batches.

**Majority-Vote DPO (MV-DPO)** follows the standard DPO formulation with preference pairs constructed from our pre-generated debate outcomes:

$$\mathcal{L}_{\text{DPO}}(\theta) = -\mathbb{E}_{x \sim \mathcal{D}} \mathbb{E}_{(y^+, y^-) \in \mathcal{G}^+(x) \times \mathcal{G}^-(x)} \left[ \log \sigma \left( \beta \left[ \log \frac{\pi_\theta(y^+|x)}{\pi_{\text{ref}}(y^+|x)} - \log \frac{\pi_\theta(y^-|x)}{\pi_{\text{ref}}(y^-|x)} \right] \right) \right]. \quad (5)$$

By contrasting the model's own consensus and dissenting trajectories, DPO's log probability ratios capture differences across entire reasoning chains, not just final answers, allowing each token to contribute to the preference signal.

**Majority-Vote KTO (MV-KTO)** applies KTO's unpaired formulation with debate-derived labels from our fixed dataset, with class-balancing weights $\lambda_+$ and $\lambda_-$:

$$\mathcal{L}_{\text{KTO}}(\theta) = - \lambda_+ \mathbb{E}_{x \sim \mathcal{D}} \mathbb{E}_{y^+ \in \mathcal{G}^+(x)} \left[ \log \sigma \left( \beta \log \frac{\pi_\theta(y^+|x)}{\pi_{\text{ref}}(y^+|x)} \right) \right]$$
$$- \lambda_- \mathbb{E}_{x \sim \mathcal{D}} \mathbb{E}_{y^- \in \mathcal{G}^-(x)} \left[ \log \sigma \left( -\beta \log \frac{\pi_\theta(y^-|x)}{\pi_{\text{ref}}(y^-|x)} \right) \right]. \quad (6)$$

KTO's unpaired structure handles imbalanced outcomes where majority trajectories dominate.

**MACA as a generalization of self-guided majority-vote RL.** MACA subsumes prior majority-vote RL approaches such as TTRL (Zuo et al., 2025) and ScPO (Prasad et al., 2024), which use a single round majority vote from independently sampled responses. TTRL uses this to inform GRPO labels, while ScPO uses it to construct DPO pairs. By contrast, MACA introduces multi-agent, multi-round debate, including deliberation traces as conditioning contexts within the prompts during RL. This exposes the model to concrete examples of relative grounding, instances where consensus emerges through explicit engagement with peer arguments, allowing the model to learn not just to aggregate outputs, but to internalize the deliberative process by which consensus is reached.

## 5 RESULTS AND DISCUSSION

We evaluate MACA by post-training four instruction-tuned small LMs (Qwen-2B (Yang et al., 2024), Llama-3B (Grattafiori et al., 2024), Phi-4B (Abdin et al., 2024), and Llama-8B (Grattafiori et al., 2024)) on six reasoning benchmarks (MATH (Hendrycks et al., 2021), GSM8K (Cobbe et al., 2021), MathQA (Amini et al., 2019), SVAMP (Patel et al., 2021), GPQA (Rein et al., 2023), and Commonsense-QA (Talmor et al., 2019)). We use 4-bit quantization with QLoRA (Dettmers et al., 2023) and limit responses to 256 tokens with temperature $\tau = 1.0$, which tests exploratory sampling under a budget adequate for efficient solvers on these benchmarks while reflecting realistic deployment compute constraints (Tong et al., 2025; Husom et al., 2025). Improvements persist when tested with larger token limits (App. M). We instantiate multi-agent debate with $M = 3$ clones and $R = 2$ rounds, and compare (1) pre-trained models, (2) SFT baselines, and (3) MACA variants of GRPO, DPO, and KTO. We train and evaluate models on 1500/500 train/test splits for each dataset independently, unless otherwise specified, isolating task-specific self-consistency improvements. We report mean agent accuracy with standard deviation across three seeds. Debate prompts, training parameters, multi-processing design, and other experimental details can be found in App. E.

### 5.1 POST-TRAINING IMPROVES SELF-CONSISTENCY

We measure the effect of post-training on sampling consistency $s_t^{\theta,\tau}(x)$, the fraction of sampled trajectories that match the majority answer. As formalized in Sec. 3, we track the sampling consistency where $s_t^{\theta,\tau}(x)$ converges to the modal probability $S_{\theta,\tau}^+(x)$ as $t \to \infty$. For each model, we sample 20 trajectories on 500 held-out prompts and evaluate $s_t^{\theta,\tau}(x)$ for $t = 1$ to 20 (Fig. 2). At $t = 1$, this metric primarily captures answer completeness, i.e., the percentage of responses that produce parseable answers within the token window, which post-training substantially improves. MV-DPO and MV-KTO achieve these gains through self-supervised preference learning alone, demonstrating that models can teach themselves more efficient reasoning without format rewards (App. M), which can otherwise be spurious (Huang et al., 2024; Srivastava et al., 2025). As sample size increases, the metric transitions to measuring true cross-sample agreement. The curves stabilize up to 27.6 percentage points above baseline, demonstrating that post-training increases answer concentration: models more consistently sample trajectories that converge despite high-temperature exploration. These improvements persist when tested without max token constraints (App. J.1).

In multi-agent debate, we evaluate the agreement metric $d_M^{\theta,\tau}(x)$ defined in Sec. 3. Base small LMs initially struggle to reach meaningful consensus: for Qwen-2B on GSM8K, most of the "consensus" comes from random tie-breaking (1/3 agreement) or weak majorities (2/3), with only 13.4% reaching unanimity (Fig. 12, App. P). Post-training with MACA thoroughly addresses this: non-parseable responses drop from 13.8% to 0.6%, no-agreement cases from 45.6% to 19.8%, while unanimous agreement triples from 13.4% to 43.4%. This confirms that MACA improves both individual reasoning quality and collaborative grounding, enabling genuine consensus rather than noisy aggregation (agreement distributions in App. P).

### 5.2 SELF-CONSISTENCY IMPROVES PROBLEM-SOLVING PERFORMANCE

**Impact on a single agent in zero-shot setting.** Across 12 model-dataset pairs, MV-RL methods consistently outperform the Base and MV-SFT baselines (Table 1) in single-agent zero-shot (single trajectory) settings. Self-guided preference learning (MV-DPO and MV-KTO) outperforms scalar rewards via MV-GRPO for all models except Phi-4B. MV-DPO is best in 7/12 cases, while MV-KTO is better in some cases for smaller LMs.

**Impact on inference-time sampling.** We next examine how gains in self-consistency translate to inference-time performance under various sampling regimes. For each prompt, we draw $t$ trajectories and report two metrics: Pass@t, the fraction of prompts for which at least one of the first $t$ samples is correct (Chen et al., 2021), and MV@t, the fraction for which a majority vote over those $t$ samples matches the ground truth (ties counted as incorrect) (Li et al., 2024). As shown in Fig. 3, post-training (blue) (i) lifts greedy ($\tau$=0) accuracy, (ii) increases MV@t at each fixed $t$, and (iii) raises Pass@t, the empirical sampling upper bound, indicating higher achievable accuracy at any given sampling budget. When additional inference compute is available (i.e., $t > 1$), sampling techniques continue to see gains on top of the post-training improvements, showing MACA

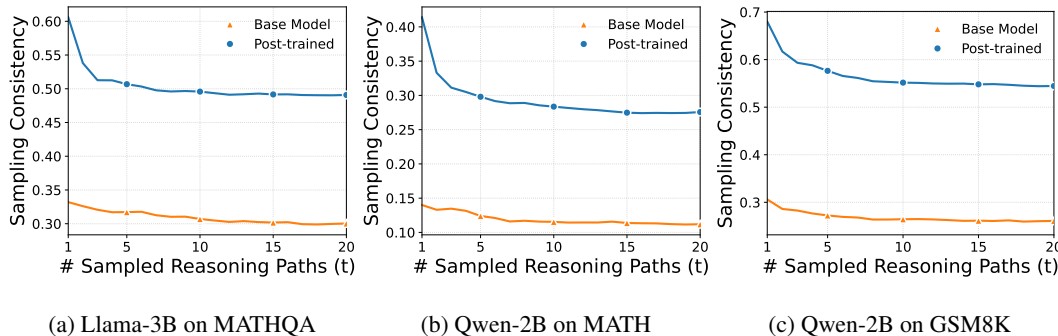

(a) Llama-3B on MATHQA      (b) Qwen-2B on MATH      (c) Qwen-2B on GSM8K

Figure 2: Consistency before and after MACA post-training. Pre-trained models (Orange) show low sampling consistency across sampled trajectories. Post-training with MACA (Blue) substantially improves sampling consistency. Averaged over 500 test prompts with 20 trajectories each.

Table 1: Accuracy impact of MACA on single agent performance in zero-shot setting.

| Model | Dataset | Base / SFT (Baseline) | | RL (Our Methods) | | | Best Δ |
|-------|---------|------|---------|---------|--------|--------|--------|
| | | Base | MV-SFT | MV-GRPO | MV-KTO | MV-DPO | |
| Qwen2B | MATH | 7.67 | $11.51 \pm 0.60$ | $18.09 \pm 0.71$ | $20.18 \pm 0.67$ | $\mathbf{23.49 \pm 2.30}$ | ↑15.82 |
| | GSM8K | 23.00 | $24.84 \pm 0.87$ | $34.40 \pm 2.08$ | $\mathbf{45.13 \pm 1.80}$ | $43.87 \pm 1.92$ | ↑22.71 |
| | MathQA | 5.00 | $5.25 \pm 0.33$ | $17.27 \pm 2.88$ | $\mathbf{22.16 \pm 1.14}$ | $20.91 \pm 0.47$ | ↑17.27 |
| Llama3B | MATH | 27.87 | $25.89 \pm 0.56$ | $35.22 \pm 0.44$ | $40.64 \pm 1.25$ | $\mathbf{40.71 \pm 0.08}$ | ↑13.26 |
| | GSM8K | 57.33 | $55.98 \pm 0.68$ | $52.40 \pm 2.84$ | $\mathbf{65.76 \pm 1.44}$ | $64.98 \pm 1.67$ | ↑8.80 |
| | MathQA | 23.87 | $23.44 \pm 0.73$ | $30.09 \pm 1.98$ | $42.84 \pm 0.67$ | $\mathbf{45.00 \pm 2.23}$ | ↑21.13 |
| Phi4B | MATH | 34.60 | $34.60 \pm 0.82$ | $\mathbf{37.42 \pm 0.16}$ | $33.84 \pm 0.78$ | $34.62 \pm 1.48$ | ↑2.82 |
| | GSM8K | 67.27 | $69.58 \pm 0.76$ | $67.13 \pm 3.60$ | $75.60 \pm 1.80$ | $\mathbf{76.87 \pm 0.36}$ | ↑9.84 |
| | MathQA | 34.87 | $34.04 \pm 0.58$ | $\mathbf{45.52 \pm 2.19}$ | $33.91 \pm 0.16$ | $33.91 \pm 0.50$ | ↑10.65 |
| Llama8B | MATH | 22.93 | $23.16 \pm 0.14$ | $29.66 \pm 1.27$ | $39.42 \pm 0.44$ | $\mathbf{46.00 \pm 0.35}$ | ↑23.07 |
| | GSM8K | 57.93 | $42.09 \pm 1.28$ | $62.45 \pm 6.01$ | $72.36 \pm 1.34$ | $\mathbf{77.36 \pm 0.27}$ | ↑19.43 |
| | MathQA | 29.67 | $30.84 \pm 0.60$ | $33.07 \pm 1.11$ | $38.42 \pm 1.22$ | $\mathbf{51.18 \pm 0.24}$ | ↑21.51 |

is complementary to inference-time sampling. The same pattern holds with full-precision, with improvements comparable to their 4-bit counterparts (App. O). Finally, self-consistency gains strongly correlate with accuracy improvements ($r > 0.86$ across all tested inference conditions; see App. J).

**Impact on multi-agent debate setting.** We evaluate MACA in the multi-agent setting using final-round majority-vote accuracy (the consensus after debate). Baselines are (1) the base model's debate outcome (Du et al., 2023) and (2) MV-SFT on majority traces (Subramaniam et al., 2025). As shown in Table 2, post-training on debate-derived signals improves ensemble accuracy across all models and datasets. Preference learning variants (MV-DPO and MV-KTO) provide the largest and most consistent gains, up to +42.73 percentage points, by directly optimizing the log-probability gap between trajectories, with MV-KTO performing best on smaller models ($\leq 3B$) and MV-DPO on larger ones ($4-8B$). Learning the relative separation between full reasoning trajectory pairs appears to better address credit assignment challenges in sparse final-answer supervision compared to both MV-GRPO's scalar rewards and MV-SFT's imitation learning, simultaneously improving answer accuracy while reinforcing higher-quality and more concise intermediate reasoning steps (App. M).

**Impact on debate dynamics.** Fig. 4 summarizes four metrics: initial round average single-agent accuracy, initial round ensemble majority vote (not used in the debate), final round average single-agent accuracy (conditioned on debate context/peer chains-of-thought), and final round ensemble majority vote (consensus). Post-training produces the largest gains where agents leverage peer feedback, since it directly teaches effective peer context utilization. We also observe that post-training mitigates debate-driven performance degradation observed in MATHQA for Llama-3B. The con-

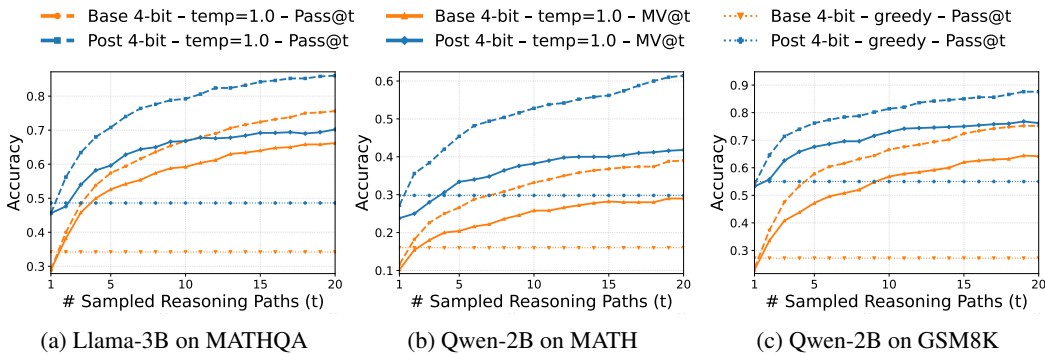

(a) Llama-3B on MATHQA      (b) Qwen-2B on MATH      (c) Qwen-2B on GSM8K

Figure 3: Post-training self-consistency improves sampling accuracy. Dashed: Pass@t (oracle upper bound), solid: MV@t (majority over $t$ samples), dotted: greedy ($\tau = 0$) accuracy. (Blue): post-trained model. (Orange): base model. Curves computed over 500 prompts.

Table 2: Post-training multi-agent debate yields consistent accuracy improvements.

| Model | Dataset | Pre / SFT (Baseline) | | RL (Our Methods) | | | Best Δ |
|-------|---------|------|---------|----------|---------|---------|--------|
| | | Debate | MV-SFT | MV-GRPO | MV-KTO | MV-DPO | |
| Qwen2B | MATH | 32.40 | 37.07 ± 3.07 | 39.00 ± 1.74 | **46.47 ± 3.01** | 42.60 ± 1.78 | ↑14.07 |
| | GSM8K | 49.60 | 50.53 ± 1.36 | 54.13 ± 2.02 | **63.07 ± 0.64** | 58.47 ± 1.62 | ↑13.47 |
| | MathQA | 24.20 | 26.27 ± 0.58 | 29.93 ± 4.99 | **32.60 ± 0.72** | 28.33 ± 0.31 | ↑9.13 |
| Llama3B | MATH | 37.80 | 35.33 ± 1.62 | 48.33 ± 2.19 | **52.93 ± 0.99** | 51.93 ± 1.67 | ↑15.27 |
| | GSM8K | 65.60 | 64.60 ± 1.59 | 68.60 ± 1.00 | **73.13 ± 0.83** | 71.67 ± 3.03 | ↑7.80 |
| | MathQA | 21.60 | 40.07 ± 2.00 | 48.73 ± 1.60 | **64.00 ± 0.53** | 63.13 ± 1.89 | ↑42.73 |
| Phi4B | MATH | 44.40 | 45.53 ± 2.53 | **49.93 ± 1.33** | 45.27 ± 0.70 | 46.73 ± 1.67 | ↑5.53 |
| | GSM8K | 79.60 | 78.93 ± 0.61 | 82.67 ± 1.81 | 82.47 ± 1.14 | **84.73 ± 0.31** | ↑5.13 |
| | MathQA | 49.60 | 50.87 ± 1.62 | **63.07 ± 1.21** | 51.53 ± 1.14 | 51.40 ± 1.44 | ↑13.47 |
| Llama8B | MATH | 32.80 | 34.13 ± 0.70 | 45.93 ± 1.03 | 53.93 ± 1.80 | **59.67 ± 1.33** | ↑26.87 |
| | GSM8K | 74.00 | 66.27 ± 1.01 | 81.53 ± 2.81 | 81.00 ± 1.97 | **81.93 ± 1.51** | ↑8.60 |
| | MathQA | 44.60 | 44.13 ± 1.10 | 57.27 ± 0.61 | 62.00 ± 2.03 | **69.27 ± 1.55** | ↑24.67 |

trastive signal allows models to learn from consensus patterns even when exposed to flawed reasoning trajectories, correcting rather than amplifying poor grounding behaviors.

**Improving self-consistency on math datasets improves general reasoning.** We demonstrate that training to improve internal self-consistency on any mathematical dataset enhances performance across diverse reasoning tasks, including previously unseen domains. Building on recent work

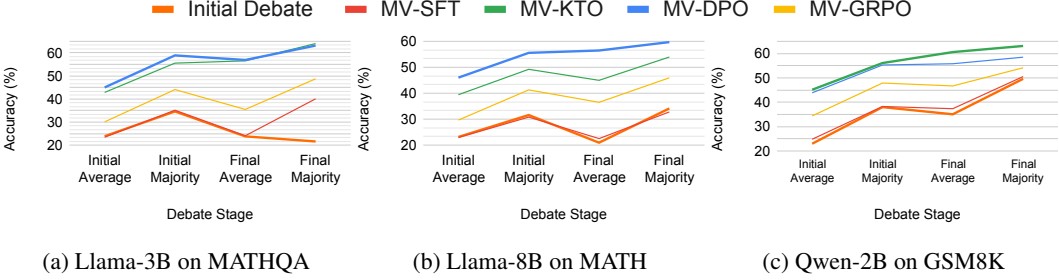

(a) Llama-3B on MATHQA      (b) Llama-8B on MATH      (c) Qwen-2B on GSM8K

Figure 4: Debate-aware RL improves all stages of multi-agent debate. Incorporating debate context in RL teaches agents to leverage prior arguments, improving final-round consensus. Stages: initial round average, initial round majority vote, final round average, final round majority vote.

Table 3: Post-training self-consistency improves performance across general reasoning benchmarks. Models trained on datasets (columns) tested on benchmarks (rows). Bottom rows show generalization to unseen benchmarks: SVAMP (math), GPQA (science), CSQA (commonsense). All = joint training on combined datasets. Arrows show absolute gains over instruction-tuned model.

| Test | Qwen2B (post-trained on) | | | | | Llama3B (post-trained on) | | | | |
|------|------|------|------|------|------|------|------|------|------|------|
| | Base | MATH | GSM | MQA | All | Base | MATH | GSM | MQA | All |
| MATH | 10.4 | ↑10.0 | ↑3.8 | ↑10.8 | ↑12.2 | 32.0 | ↑19.4 | ↑16.4 | ↑18.2 | ↑21.2 |
| GSM | 27.0 | ↑20.0 | ↑25.6 | ↑22.6 | ↑27.8 | 69.6 | ↑6.0 | ↑6.8 | ↑8.4 | ↑10.8 |
| MQA | 7.4 | ↑12.6 | ↑17.0 | ↑15.4 | ↑21.4 | 24.6 | ↑14.0 | ↑13.4 | ↑21.2 | ↑21.6 |
| SVAMP | 48.3 | ↑19.0 | ↑18.0 | ↑17.0 | ↑27.7 | 71.3 | ↑6.0 | ↑6.4 | ↑9.7 | ↑7.1 |
| GPQA | 0.5 | ↑6.0 | ↑5.3 | ↑12.8 | ↑16.3 | 0.7 | ↑5.4 | ↑6.3 | ↑9.8 | ↑10.7 |
| CSQA | 3.8 | ↑19.8 | ↑43.0 | ↑54.0 | ↑59.6 | 53.0 | ↑7.4 | ↑10.6 | ↑11.6 | ↑11.0 |

Table 4: Multi-agent: Post-training with debate (MV) is comparable to ground-truth (GT).

| Llama-8B | Dataset | Debate | SFT | | KTO | | DPO | | GRPO | |
|------|------|------|------|------|------|------|------|------|------|------|
| | | | *GT* | *MV* | *GT* | *MV* | *GT* | *MV* | *GT* | *MV* |
| Single-Agent | MATH | 22.93 | **23.73** | 23.00 | **41.20** | 39.40 | 45.13 | **46.40** | 29.07 | **31.13** |
| | GSM8K | 57.93 | 40.20 | **41.67** | **72.60** | 70.87 | 76.33 | **77.67** | 61.27 | **66.87** |
| Multi-Agent | MATH | 32.80 | **34.00** | 33.40 | **56.00** | 55.80 | **61.80** | 60.80 | **48.60** | 44.80 |
| | GSM8K | 74.00 | **65.20** | **65.20** | **81.20** | 79.40 | 81.60 | **83.0** | 83.20 | **84.20** |

showing math training enhances general reasoning (DeepSeek-AI, 2025; Akter et al., 2025), we demonstrate that improving self-consistency—internal consensus strength—on math is essential for reliably unlocking these capabilities. Table 3 reports results for models trained on MATH, GSM8K, or MathQA individually, and on all three combined (All), using MV-DPO. Training on any single dataset improves performance across all reasoning tasks, including unseen math (SVAMP), science (GPQA), and commonsense reasoning (CSQA). Joint training achieves further improvements across nearly every benchmark, demonstrating that diverse training data amplifies self-consistency gains.

## 5.3 ABLATION STUDY

To understand the sources of performance gains from MACA, we conduct ablation studies examining key components. We show that self-generated consensus signals outperform ground-truth supervision, including peer context during training improves relative grounding and debate utilization, and multi-round debate provides stronger signals than single-round majority vote. We also demonstrate in App. M that self-supervised preference learning serves as an effective implicit format reward by reinforcing more efficient and concise chain-of-thought construction, though most gains stem from fundamental reasoning improvements rather than formatting alone.

**RL with debate is comparable to RL with ground-truth.** Table 4 compares post-training with debate majority-vote labels, derived from the model's own consensus without external supervision, to ground-truth labels on Llama-3B. Across post-training methods and in both single- and multi-agent settings, using Debate-MV is consistently comparable to ground-truth supervision, a trend that holds across other models as well (App. F). We additionally show in App. K that this comparable performance between Debate-MV and ground-truth supervision extends to general reasoning improvements under MACA across all proposed post-training methods and reasoning benchmarks tested, as observed in Table 3. Because Debate-MV scales naturally with sample size, its parity with ground-truth highlights self-supervised alignment as a promising direction.

**RL with debate outperforms RL with single-round majority vote.** We evaluate three training conditions in the single-agent setting to understand the impact of training on the debate versus initial round majority vote on the individual agents themselves, as shown in Table 6: (1) initial round majority vote with GRPO (akin to TTRL (Zuo et al., 2025)), (2) initial round majority vote with DPO (akin to ScPO without the weighted loss term (Prasad et al., 2024)), and (3) debate majority

Table 5: Multi-agent: Post-training with debate context (peer CoTs) outperforms no context.

| GSM8K | Initial Round Avg | | Final Round Avg | | Final Round MV | |
|---|---|---|---|---|---|---|
| | NoCtx | Context | NoCtx | Context | NoCtx | Context |
| Llama-3B | 65.87 | **66.73** | 67.87 | **70.60** | 70.60 | **73.80** |
| Phi-4B | 74.67 | **76.60** | 74.60 | **77.80** | 81.80 | **84.40** |
| Llama-8B | 75.80 | **77.67** | 78.00 | **81.20** | 81.60 | **83.00** |

Table 6: Single-agent: RL on debate majority-vote with peer context (MV-DPO) outperforms RL on single-round majority vote (TTRL, ScPO). Arrows show gains over instruction-tuned model. Statistics gathered over 3 seeds.

| | MATH | | | | GSM8K | | | |
|---|---|---|---|---|---|---|---|---|
| | Qwen-2B | Llama-3B | Phi-4B | Llama-8B | Qwen-2B | Llama-3B | Phi-4B | Llama-8B |
| TTRL | $\uparrow$**18.0**$_{\pm 2.9}$ | $\uparrow$5.3$_{\pm 5.7}$ | $\uparrow$6.1$_{\pm 2.1}$ | $\uparrow$7.5$_{\pm 0.2}$ | $\uparrow$**24.5**$_{\pm 1.6}$ | $\downarrow$22.3$_{\pm 8.3}$ | $\uparrow$4.8$_{\pm 1.6}$ | $\downarrow$13.5$_{\pm 5.7}$ |
| ScPO | $\uparrow$2.3$_{\pm 1.1}$ | $\uparrow$3.4$_{\pm 0.2}$ | $\uparrow$0.1$_{\pm 0.5}$ | $\uparrow$3.7$_{\pm 0.6}$ | $\uparrow$15.9$_{\pm 3.4}$ | $\uparrow$1.6$_{\pm 0.8}$ | $\uparrow$5.9$_{\pm 0.5}$ | $\uparrow$12.9$_{\pm 1.2}$ |
| MV-DPO | $\uparrow$16.7$_{\pm 0.4}$ | $\uparrow$**12.5**$_{\pm 0.7}$ | $\uparrow$**6.9**$_{\pm 0.2}$ | $\uparrow$**17.1**$_{\pm 0.8}$ | $\uparrow$23.8$_{\pm 0.9}$ | $\uparrow$**3.7**$_{\pm 0.3}$ | $\uparrow$**7.3**$_{\pm 0.3}$ | $\uparrow$**18.2**$_{\pm 0.2}$ |

vote with DPO including peer context (MV-DPO from MACA). Training on the debate signal improves single-agent performance more than reinforcing the initial round majority vote. Additionally, Table 5 shows that including peer context during preference learning (MV-DPO) improves multi-agent debate performance (more models/datasets in App. G), with the largest gains in final-round individual accuracy, reflecting better use of peer chains-of-thought during deliberation (Fig. 4).

**Debate improves consensus quality.** Finally, the debate process itself improves consensus signals. Base small LMs produce mostly random or weak majorities initially, but post-training increases unanimous agreement from 27.2% to 43.4% between initial and final rounds (Qwen-2B on GSM8K, App. P). This iterative refinement provides more reliable training signals than independent sampling. While majority vote correlates strongly with accuracy (alternative ranking signals like log-probability are explored in App. I), post-training debate creates a self-reinforcing cycle: better debate generates higher-quality signals, producing models that debate more effectively. Iterative training yields continued improvements with diminishing returns, as shown in App. H.

## 6 CONCLUSION, LIMITATIONS, AND FUTURE WORK

We introduce self-consistency as an intrinsic property of well-aligned reasoning and present MACA, a self-supervised framework that teaches models to reliably sample coherent reasoning through reinforcement of internal consensus signals. Without external supervision, MACA drives substantial improvements: +27.6% self-consistency on GSM8K, +22.4% Pass@20 on MATH, +23.7% zero-shot accuracy on MATH, strong generalization to unseen benchmarks (+16.3% GPQA, +11.6% CommonsenseQA), and enhanced multi-agent performance (+42.7% on MathQA). Through debate, agents attempt to ground their reasoning in peer context, and we select trajectories where such deliberation aligns with internal consensus. These consensus-aligned examples provide natural supervision for stable reasoning patterns, enabling models to self-improve their reasoning consistency, efficiency, and accuracy without explicit chain-of-thought supervision or external answer verification. Through preference learning on debate signals, models produce more concise chains-of-thought without format rewards and correct cases where aggregation previously led to degeneration, significantly improving the robustness of multi-agent debate. While MACA has limitations, including requiring sufficient base model competence to generate meaningful consensus signals, potentially amplifying existing biases in model outputs, and lacking direct supervision of intermediate reasoning correctness, it demonstrates that consensus reinforcement has robust benefits. Future work could explore alternative consensus methods, confidence-weighted voting, heterogeneous agents, and better leveraging of minority traces. Sizeable gains on difficult unseen tasks (+10.0% on AMC, App. M) suggest consensus alignment unlocks latent capabilities beyond consistency itself. MACA demonstrates that language models can effectively harness internal deliberation for self-alignment, elevating their reasoning capabilities autonomously.

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

## A    LLM Usage Declaration

We declare that large language models (LLMs) were only used to assist with minor tasks, including grammar checking, language refinement, and basic coding support such as debugging. All research contributions, experimental design, data analysis, and scientific findings presented in this work are original to the authors.

## B    Examples of Post-Trained Consistent Reasoning in Single- and Multi- Agent Settings

This appendix presents examples from the debates, providing a qualitative signal of how post-training affects both single-agent reasoning and multi-agent collaboration. The evidence highlights visible improvements in mathematical reasoning, consensus formation, and systematic problem-solving.

### B.1    Overview of Key Findings

Analysis of the transcripts reveals that post-training leads to more systematic and efficient reasoning in single-agent settings. Models demonstrate clearer step-by-step problem decomposition, make fewer calculation errors, and show improved conceptual understanding. Notably, responses become more concise and are less likely to be truncated due to inefficient reasoning.

In multi-agent debates, post-training enhances collaborative effects. Agents more frequently reference and build upon each other's solutions, leading to more robust cross-verification and consensus formation. We observe that post-trained agents are more likely to resolve mistakes through discussion rather than propagate errors.

Our analysis shows that preference learning teaches models higher-quality and more concise reasoning without requiring explicit format rewards. The consensus mechanism itself becomes more effective at aggregating individual outputs, supporting both in-distribution and out-of-distribution transfer.

### B.2    Case Study 1: Single-Agent Mathematical Reasoning

### B.2.1    Problem: Cities A and B Relative Speed

> **AMC Problem Statement**
>
> Cities $A$ and $B$ are 45 miles apart. Alicia lives in $A$ and Beth lives in $B$. Alicia bikes towards $B$ at 18 miles per hour. Leaving at the same time, Beth bikes toward $A$ at 12 miles per hour. How many miles from City $A$ will they be when they meet?
> **Correct Solution:** Combined speed = 18 + 12 = 30 mph; Time to meet = 45/30 = 1.5 hours; Distance from A = 18 × 1.5 = 27 miles
> **Ground Truth:** 27

### B.2.2 PRE-TRAINING RESPONSE: FUNDAMENTAL CONCEPTUAL CONFUSION

**Base Model: Conceptual Breakdown**

To solve this problem, we can use the concept of relative motion. Here's a step-by-step breakdown:

- Let's denote the time that Alicia travels as $t$ hours. Since Beth travels in the opposite direction, we can say that the time she travels is also $t$ hours.
- Since they are traveling towards each other, the distance between them will decrease at a rate of $18 + 12 = 30$ miles per hour.
- The initial distance between them is 45 miles. After $t$ hours, the total distance traveled by both Alicia and Beth will be the initial distance plus the distance traveled together, which is $45 + 30t$ miles.
- We can set up an equation using the fact that their combined distance traveled equals the initial distance between them: $45 + 30t = 45$. This is because they meet when the sum of their distances is equal to the initial distance.
- Solving for $t$, we get $t = 0$ hours. However, this doesn't make sense because it means they meet immediately after starting. Let's try again.
- Since we can't set up a simple equation with this method, let's think about it in terms of their relative speed: 12

[Response cuts off mid-sentence]
**Parsed Answer:** null

**Analysis: Multiple failure modes.** The pre-training response reveals several critical reasoning failures: incorrect problem setup, misapplication of distance-time relationships, and ultimately an illogical result. The model exhausts its token budget exploring unproductive solution paths before reaching a conclusion.

### B.2.3 POST-TRAINING RESPONSE: CORRECT SYSTEMATIC SOLUTION

**Post-Training: Correct and Concise**

To solve this problem, we can follow these steps:

- We know the distance between Cities A and B is 45 miles.
- Alicia and Beth are biking towards each other at a combined speed of $18 + 12 = 30$ miles per hour.
- Since they are biking towards each other, we can use the concept of relative speed to find the time it takes for them to meet.
- We can calculate the time using the formula: time = distance / speed. Using this formula, the time they take to meet is 45 miles / 30 miles per hour, which equals 1.5 hours.
- Now that we have the time, we can find the distance covered by Alicia, since we know her speed is 18 miles per hour. Distance covered by Alicia is $18 \times 1.5$, which equals 27 miles.

Therefore, Alicia and Beth will meet 27 miles from City A.

27

**Analysis: Systematic improvement.** Post-training produces a clear, systematic approach: correct conceptual application, logical step progression, and efficient completion within the token limit. This demonstrates improved reasoning quality and token efficiency achieved through preference learning without explicit format rewards.

### B.3 CASE STUDY 2: MULTI-AGENT MATHEMATICAL REASONING

### B.3.1 PROBLEM: ROAD TRIP GAS MILEAGE CALCULATION

---

**Problem Statement**

Sophia is thinking of taking a road trip in her car, and would like to know how far she can drive on a single tank of gas. She has traveled 100 miles since last filling her tank, and she needed to put in 4 gallons of gas to fill it up again. The owner's manual for her car says that her tank holds 12 gallons of gas. How many miles can Sophia drive on a single tank of gas?
**Correct Solution:** $100 \div 4 = 25$ mpg; $25 \times 12 = 300$ miles
**Ground Truth:** 300

---

### B.4 COMPLETE MULTI-AGENT TRANSCRIPTS: PRE-TRAINING FAILURE (ITERATION 0)

### B.4.1 ROUND 1: INITIAL AGENT RESPONSES

---

**User Prompt**

Solve the following math problem. Sophia is thinking of taking a road trip in her car, and would like to know how far she can drive on a single tank of gas. She has traveled 100 miles since last filling her tank, and she needed to put in 4 gallons of gas to fill it up again. The owner's manual for her car says that her tank holds 12 gallons of gas. How many miles can Sophia drive on a single tank of gas?
Provide a bullet point summary of your step-by-step reasoning. Your final answer should be a single numerical number, in the form \boxed{answer}, at the end of your response.

---

**Agent 1: Mathematical Error**

To solve this problem, I need to determine Sophia's car's fuel efficiency and then calculate how far she can drive on a full tank.
Step 1: Determine fuel efficiency from the given information.

- Sophia traveled 100 miles and used 4 gallons of gas
- Fuel efficiency = 100 miles ÷ 4 gallons = 25 miles per gallon

Step 5: Convert the fuel efficiency from mpg to gallons per mile (GPM):

- Since 1 mpg means traveling 1 mile with each gallon, Sophia will use 4 gallons × (1 mile/mpg) = 4 miles per tank.

Step 6: Calculate total distance for full tank:

- Total distance = 4 miles per tank × 12 gallons = 48 miles

$\boxed{48}$

---

**Agent 2: Incomplete Reasoning**

If the distance covered in one tank of fuel is 100 miles, then the Miles Per Gallon (MPG) is calculated as follows...
[Response appears incomplete and truncated mid-calculation]
After reviewing the problem again, I need to be more careful with my calculations.
Let me reconsider: Sophia used 4 gallons to travel some distance, and her tank holds 12 gallons total.
I'm getting confused by the different approaches. Let me try again: 100 miles with 4 gallons means...
Final answer: $\boxed{48}$

---

> **Agent 3: Correct but Overruled**
>
> Let me solve this step by step:
> Step 1: Calculate fuel efficiency from given data:
> - Distance traveled: 100 miles
> - Gas used: 4 gallons
> - Fuel efficiency: $100 \div 4 = 25$ miles per gallon
>
> Step 2: Calculate range on full tank:
> - Full tank capacity: 12 gallons
> - Total range: 25 mpg × 12 gallons = 300 miles
>
> Therefore, Sophia can drive 300 miles on a single tank of gas.
> 300

### B.4.2 ROUND 2: FAILED COLLABORATIVE DISCUSSION

> **Agent 1: Doubling Down on Error**
>
> After reviewing all approaches, I maintain my calculation:
> Based on the calculations, I'm confident that 48 miles is correct because we need to convert the efficiency properly.
> 48

**Pre-Training Result:** Consensus selected 48 miles despite Agent 3's correct reasoning.

### B.4.3 COMPLETE MULTI-AGENT TRANSCRIPTS: POST-TRAINING SUCCESS (ITERATION 1)

### B.4.4 ROUND 1: PERFECT AGENT RESPONSES

> **Agent 1: Complete Correction**
>
> To solve this problem, I need to determine Sophia's fuel efficiency and calculate her range on a full tank.
> Step 1: Calculate Sophia's car's fuel efficiency:
> - Sophia traveled 100 miles since last fill-up
> - She used 4 gallons of gas
> - Fuel efficiency = 100 miles ÷ 4 gallons = 25 miles per gallon
>
> Step 2: Calculate range on full tank:
> - Tank capacity: 12 gallons
> - Range = 25 mpg × 12 gallons = 300 miles
>
> Therefore, Sophia can drive 300 miles on a single tank of gas.
> 300

> **Unanimous Agreement**
>
> **Agent 1:** "I see all agents used the same systematic approach. The calculation is clear: 25 mpg × 12 gallons = 300 miles."
> **Agent 2:** "Perfect agreement on methodology. The fuel efficiency calculation is the key step, then applying it to full tank capacity."
> **Agent 3:** "Unanimous consensus on 300 miles. All reasoning aligns perfectly."

**Post-Training Result:** Perfect consensus on 300 miles with all agents showing correct reasoning.

### B.4.5 ANALYSIS: TRAINING EFFECTS ON MULTI-AGENT COLLABORATION

Post-training produces substantial improvements in both individual and collective reasoning. Before training, agents make calculation errors and the consensus mechanism selects incorrect answers even when correct solutions are present, with two agents converging on the same mistake and overruling the correct agent. After training, all agents apply correct methodology and reach unanimous agreement on the right answer. These results demonstrate that preference learning encourages systematic step-by-step reasoning, reducing errors and their propagation during collaboration, while multi-agent training enhances the consensus mechanism's ability to reliably select correct solutions.

## C GROUNDING BEHAVIORS IN BASE-MODEL MULTI-AGENT DEBATES

Having provided examples of the impact of post-training on single- and multi-agent reasoning, we now highlight representative examples from some initial debates among base models demonstrating spontaneous grounding behaviors that emerge during multi-agent reasoning. Note that each debate consists of an initial round (three agents provide independent solutions) and a final round (agents update after seeing peers' responses).

### C.1 EXAMPLE 1: AIRPORT TIMING PROBLEM

**Problem**

Amber and Josh are flying to France for vacation. Their flight leaves at 8:00 pm and as an international flight, it's recommended that they check in 2 hours early. It will take 45 minutes to drive to the airport and another 15 minutes to park their vehicle and make their way to the terminal. What is the latest time they can leave their house to make it to the airport in time?

The correct solution requires working backwards from the 8:00 PM flight time: they must arrive by 6:00 PM (2 hours early), and with 1 hour total travel time, they must leave by 5:00 PM.

**Initial round:**

**Agent 1: Directional Error**

- Flight leaves at 8:00 pm, need to check in 2 hours early = 6:00 pm arrival
- Calculate travel time: 45 + 15 = 60 minutes = 1 hour
- Add travel time to departure time: 6:00 pm + 1 hour = 7:00 pm

Error: Adds travel time instead of subtracting it
Answer: 7:00 pm

**Agent 2: Fundamental Misunderstanding**

- Find total travel time: 45 + 15 = 60 minutes = 1 hour
- Add travel time to flight departure time: 8:00 pm + 1 hour = 9:00 pm

Completely ignores 2-hour check-in requirement and adds travel time instead of subtracting it
Answer: 9:00 pm

**Agent 3: Correct Solution**

- Flight at 8:00 pm requires arrival by 6:00 pm (2 hours early)
- Travel time: 45 min driving + 15 min parking = 60 minutes
- Must leave by: 6:00 pm - 1 hour = 5:00 pm

Answer: 5:00 pm

**Final round (after seeing peers):**

---

**Agent 1: Adopts Worse Error**

"After reviewing other solutions, I update my approach:

- Departure time from house: 8:00 pm
- Total travel time: 45 + 15 = 60 minutes = 1 hour
- Add travel time to departure: 8:00 pm + 1 hour = 9:00 pm"

Abandons partial understanding to adopt Agent 2's incorrect reasoning as well
Answer: 9:00 pm

---

**Agent 2: Persists with Error**

Maintains original incorrect answer of 9:00 pm

---

**Agent 3: Explicitly Identifies Error Pattern**

"The other agents seem to be adding time to the flight departure rather than working backwards from the required arrival time. The correct approach:

- Required arrival: 6:00 pm (2 hours before 8:00 pm flight)
- Subtract travel time: 6:00 pm - 1 hour = 5:00 pm departure"

Recognizes error in other agents' responses and upholds its own correct understanding
Answer: 5:00 pm

---

**Key observation.** Agent 1 had partial understanding (recognized the 6:00 PM arrival requirement) but abandoned this to adopt Agent 2's more fundamental error. Agent 3 explicitly identifies the common error pattern of "adding time to flight departure" and maintains the correct backward calculation. This incidentally reflects exactly the case where statistical aggregation can be counterproductive: the inconsistencies in reasoning among the sampled trajectories outnumber the sensible agent, and this error gets propagated. Interestingly, we show that post-training is able to correct these cases despite the initial debate datasets containing examples of bad grounding.

## C.2  EXAMPLE 2: SELF-CORRECTION IN MATHEMATICAL REASONING

---

**Problem**

A store offers a 20% discount on all items. If Sarah buys a jacket originally priced at $80 and pants originally priced at $50, how much does she pay in total after the discount?

---

The correct solution: Total original price = $130, discount = 20% of $130 = $26, final price = $104.

**Initial round:**

---

**Agent 1: Calculation Error**

Original total: $80 + $50 = $130
Discount: 20% = 0.2 × $130 = $26
Final price: $130 + $26 = $156
Error: Adds discount instead of subtracting
Answer: $156

---

**Agent 2: Correct Approach**

Total: $130
After 20% discount: $130 × 0.8 = $104
Answer: $104

---

**Agent 3: Misapplies Discount**

Jacket discount: 20% of $80 = $16, price = $64
Pants discount: 20% of $50 = $10
Total: $64 + $10 = $74
Error: Adds discounted jacket price to pants discount amount
Answer: $74

**Final round:**

**Agent 1: Self-Correction**

"I realize I added the discount instead of subtracting. Looking at Agent 2's approach of multiplying by 0.8 is cleaner: $130 \times 0.8 = $104"
Recognizes own error and adopts correct method
Answer: $104

**Agent 3: Fixes Calculation**

"Jacket after discount: $80 \times 0.8 = $64
Pants after discount: $50 \times 0.8 = $40
Total: $64 + $40 = $104"
Answer: $104

**Key observation.** Agent 1 explicitly acknowledges their error ("I added the discount instead of subtracting") and adopts Agent 2's more efficient method. Agent 3 corrects their calculation error after seeing consistent peer answers.

## C.3 EXAMPLE 3: GEOMETRIC PROBLEM - METHOD SIMPLIFICATION

**Problem**

A rectangle has a perimeter of 24 units and an area of 32 square units. What are the dimensions of the rectangle?

The correct solution: If dimensions are length $l$ and width $w$, then $2(l + w) = 24$ so $l + w = 12$, and $lw = 32$. Solving this system gives dimensions 8 and 4.

**Initial round:**

**Agent 1: Algebraic Approach**

Let length = $l$, width = $w$
From perimeter: $2l + 2w = 24$, so $l + w = 12$
From area: $lw = 32$
Setting up quadratic: $w(12 - w) = 32$
$12w - w^2 = 32$
$w^2 - 12w + 32 = 0$
Stops without solving the quadratic

**Agent 2: Direct Factoring**

$l + w = 12$ and $lw = 32$
Looking for factors of 32 that sum to 12: 8 and 4
Dimensions: $8 \times 4$
Answer: 8 units by 4 units

> **Agent 3: Quadratic Solution**
>
> From $w^2 - 12w + 32 = 0$
> Using quadratic formula or factoring: $(w - 8)(w - 4) = 0$
> So $w = 8$ or $w = 4$
> Dimensions: $8 \times 4$
> Answer: 8 units by 4 units

**Final round:**

> **Agent 1: Adopts Simpler Method**
>
> "Agent 2's approach of finding factors of 32 that sum to 12 is much more direct than solving the quadratic. Since $8 \times 4 = 32$ and $8 + 4 = 12$, the dimensions are 8 by 4."
> Recognizes and adopts more efficient solution method
> Answer: 8 units by 4 units

**Key observation.** Agent 1 explicitly acknowledges that Agent 2's factor-finding approach is "much more direct" than the algebraic method, demonstrating recognition of solution efficiency. Post-training reinforces these behaviors, enabling the model to learn patterns of concise and efficient reasoning.

### C.4 SUMMARY

These examples demonstrate observable grounding behaviors in multi-agent debates:

**Error propagation:** Agents sometimes abandon partially correct reasoning to adopt peer errors (Example 1, Agent 1).

**Explicit error recognition:** Agents identify specific errors in their own reasoning ("I added the discount instead of subtracting") and in peers' approaches ("adding time to flight departure").

**Method adoption:** Agents recognize more efficient solution methods from peers and explicitly acknowledge this ("Agent 2's approach... is much more direct").

**Self-correction through peer comparison:** Seeing consistent peer answers prompts agents to revisit and correct their calculations.

These behaviors generate training signals beyond simple majority voting. The debate format captures not just correct answers but also common error patterns, self-correction mechanisms, and explicit reasoning about solution quality, providing rich supervision for teaching models to recognize and adopt sound reasoning strategies.

## D  MACA ALGORITHM: ADDITIONAL IMPLEMENTATION DETAILS

### D.1 AGENT HETEROGENEITY AND SPECIALIZATION

In our current experiments, we use homogeneous agents (clones) that update independently over a single training iteration, enabling evaluation of both individual and averaged performance to measure the benefits of divergence. We also explore the impact of training over multiple such iterations of debate and post-training in App. H. For multi-iteration runs, agent checkpoints can be managed in two ways: agents may either remain synchronized by resetting checkpoints to the best performing one after every iteration, or diverge by updating independently from distinct trajectory data, allowing specialization and diversity through differential learning. Additionally, the MACA framework also supports heterogeneous agents, allowing different language models to be independently optimized through this framework, where agents represent distinct models or architectures from the outset, each bringing inherently different capabilities or perspectives.

---

**Algorithm 1** Multi-Agent Consensus Alignment Via Debate-RL

---

1: **input:** agents $\{\pi_{\theta_m}\}_{m=1}^M$, rounds $R$, iters $L$, batch size $B$, method in MV-SFT, MV-GRPO, MV-DPO, or MV-KTO
2: **for** $\ell = 1, \ldots, L$ **do**
3:      sample batch $\{x^{(\ell,i)}\}_{i=1}^B$ of prompts from $\mathcal{D}$
4:      **for** $i = 1, \ldots, B$ **do**                              ▷ Generate debate trajectories
5:          $y_{m,1}^{(\ell,i)} \sim \pi_{\theta_m}(\cdot \mid x^{(\ell,i)})$    $\forall m$
6:          **for** $r = 2, \ldots, R$ **do**                   ▷ Condition on peers' previous round outputs
7:              $x_{m,r}^{(\ell,i)} = [x^{(\ell,i)}; \{y_{j,r-1}^{(\ell,i)}\}_{j\neq m}]$
8:              $y_{m,r}^{(\ell,i)} \sim \pi_{\theta_m}(\cdot \mid x_{m,r}^{(\ell,i)})$    $\forall m$
9:          **end for**
10:         Parse final answers $a_m^{(\ell,i)} = \mathcal{A}(y_{m,R}^{(\ell,i)})$    $\forall m$
11:         Aggregate consensus $\hat{a}(x^{(\ell,i)}) = \text{Majority}\{a_1^{(\ell,i)}, \ldots, a_M^{(\ell,i)}\}$
12:         Extract final contexts $x_m^{(\ell,i)} \leftarrow x_{m,R}^{(\ell,i)}$ for all $m$
13:         Partition responses by consensus alignment:
             $\mathcal{G}_+^{(\ell,i)} = \{y_{m,R}^{(\ell,i)} : a_m^{(\ell,i)} = \hat{a}(x^{(\ell,i)})\}$
             $\mathcal{G}_-^{(\ell,i)} = \{y_{m,R}^{(\ell,i)} : a_m^{(\ell,i)} \neq \hat{a}(x^{(\ell,i)})\}$
14:      **end for**
15:      **build dataset:** $\mathcal{D}_{\text{post}} = \{(x^{(\ell,i)}, \hat{a}(x^{(\ell,i)}), \mathcal{G}_+^{(\ell,i)}, \mathcal{G}_-^{(\ell,i)})\}_{i=1}^B$
16:      **training data by method:**
         *MV-SFT:* for each $m$, use $\{(x_m^{(\ell,i)}, y_{m,R}^{(\ell,i)}) : y_{m,R}^{(\ell,i)} \in \mathcal{G}_+^{(\ell,i)}\}_i$.
         *MV-GRPO:* for each $m$, store $\{(x_m^{(\ell,i)}, \hat{a}(x^{(\ell,i)}))\}_i$ for reward computation on new samples.
         *MV-DPO/MV-KTO:* pool $\{(x_m^{(\ell,i)}, y_{m,R}^{(\ell,i)})\}_{i,m}$ partitioned by $\mathcal{G}^+, \mathcal{G}^-$ across agents.
17:      **update policies:** apply majority-vote objective to refine each $\pi_{\theta_m}$
18: **end for**

---

# E   EXPERIMENTAL SETUP

## E.1   TRAINING PARAMETERS

We conducted hyperparameter sweeps across learning rates from 1e-7 to 1e-5, finding that 1e-5 consistently performed best across all methods. For preference-based methods (MV-KTO, MV-DPO), we used $\beta = 0.1$ throughout all experiments. LoRA ranks and alpha values were adjusted based on model size and computational constraints, with rank/alpha of 64-128 for MV-DPO and MV-KTO, 32-64 for MV-GRPO, and 128 for MV-SFT. Tables 7–10 provide complete parameter specifications.

GSM8K, MATH, and MathQA used 1500/500/500 train/valid/test splits. SVAMP, GPQA, CSQA, and AMC were used only for testing, with 300, 448, 500, and 40 test examples, respectively.

## E.2   DATASETS

We evaluated model mathematical reasoning using seven publicly available datasets spanning a range of difficulty and subject areas: MATH (Hendrycks et al., 2021), GSM8K (Cobbe et al., 2021), MathQA (Amini et al., 2019), SVAMP (Patel et al., 2021), GPQA (Rein et al., 2023), AMC 23 KnovelEng (2023), and CommonsenseQA/CSQA (Talmor et al., 2019).

**MATH** (Hendrycks et al., 2021): The MATH dataset contains 12,500 high school mathematics problems from algebra, geometry, combinatorics, and number theory, each requiring multi-step reasoning and precise mathematical solutions.

**GSM8K** (Cobbe et al., 2021): GSM8K is composed of 8,500 grade-school-level word problems emphasizing arithmetic and logical reasoning, with step-by-step annotated solutions for each problem.

**MathQA** (Amini et al., 2019): MathQA features over 37,000 question-answer pairs based on quantitative reasoning, requiring models to convert natural language problems into mathematical expressions and perform multi-step computation.

Table 7: Training parameters.

| Parameter | SFT | GRPO | DPO | KTO |
|---|---|---|---|---|
| Learning rate | $1 \times 10^{-5-7}$ | $1 \times 10^{-5-7}$ | $1 \times 10^{-5-7}$ | $1 \times 10^{-5-7}$ |
| Weight decay | $1 \times 10^{-2}$ | $1 \times 10^{-2}$ | $1 \times 10^{-2}$ | $1 \times 10^{-2}$ |
| Batch size | 1–8 | 8 | 1–8 | 1–8 |
| Epochs | 1–3 | 1–3 | 1–3 | 1–3 |
| Gradient accumulation steps | 2–4 | 2–4 | 2–4 | 2–4 |
| Optimizer | AdamW | AdamW | AdamW | AdamW |
| LoRA rank (r) | 8–128 | 8–64 | 8–128 | 8–128 |
| LoRA alpha | 8–128 | 8–64 | 8–128 | 8–128 |
| LoRA dropout | 0.1 | 0.1 | 0.1 | 0.1 |
| LoRA target modules | q_proj, k_proj, v_proj, o_proj | | | |
| Entropy coefficient | - | 0.01 | - | - |
| Beta (regularization) | - | - | 0.1–0.3 | 0.1–0.3 |
| Number of generations per step | - | 8 | - | - |
| Maximum sequence length | Model-dependent (2048–4096) | | | |
| Mixed precision | bf16 | | | |
| Quantization | 4-bit (BitsAndBytesConfig) | | | |

Table 8: Generation parameters.

| Parameter | Value |
|---|---|
| Temperature | 1.0 |
| Top-p sampling | 0.9 |
| Maximum new tokens | 256 |
| Do sample | True |
| Pad token ID | EOS token ID |

**SVAMP** (Patel et al., 2021): SVAMP consists of carefully reworded arithmetic word problems designed to probe model robustness and prevent reliance on annotation artifacts, focusing on genuine multi-step arithmetic reasoning.

**GPQA** (Rein et al., 2023): GPQA provides 448 expert-curated, multiple-choice questions covering graduate-level biology, physics, and chemistry, emphasizing deep, multi-stage reasoning and robust factual understanding.

**AMC 23** (KnovelEng, 2023): This dataset includes recent problems drawn from the 2023 American Mathematics Competitions (AMC), spanning algebra, combinatorics, geometry, and number theory, and is useful for assessing model performance on expert-constructed math tasks

**CommonsenseQA** (Talmor et al., 2019): CommonsenseQA is a challenging multiple-choice question-answering dataset requiring models to apply commonsense reasoning over diverse everyday scenarios. The dataset comprises 12,247 questions, each designed to probe deeper, non-trivial conceptual knowledge beyond factual recall, making it a benchmark for evaluating commonsense understanding in language models.

These datasets collectively enable a thorough evaluation of analytical reasoning capabilities in large language models.

### E.3 MULTI-AGENT DEBATE INFRASTRUCTURE

Our multi-agent debate system addresses the computational challenges of training and deploying multiple agents efficiently through specialized infrastructure for quantized training isolation and scalable debate inference.

Table 9: Multi-agent debate parameters.

| Parameter | Value |
|---|---|
| Number of agents | 3 |
| Debate rounds | 2 |
| Batch debate size | 8–24 |
| Use majority vote | True |
| Include debate context | True |
| Use async debate | True |

Table 10: Model and dataset configuration.

| Parameter | Value |
|---|---|
| Base models | Phi-4B, Qwen-2B, Llama-3B |
| Training datasets | GSM8K, MATH, MathQA |
| Test datasets | GSM8K, MATH, MathQA, SVAMP, GPQA, CSQA, AMC |
| Training size | 1500 samples |
| Test size | 500 samples |
| Validation size | 500 samples |
| Use quantization | 4-bit BitsAndBytesConfig |
| Use adapter mode | True (LoRA) |
| Use scheduler | True |

### E.3.1 QUANTIZED MARL TRAINING

Current post-training libraries lack support for multi-gpu distributed training with 4-bit quantization and PEFT adapters. We implement training isolation where each agent trains on a single GPU with explicit resource assignment, using gradient checkpointing and paged optimizers (paged_adamw_8bit) for memory efficiency. This enables stable multi-agent RL with QLoRA while preserving quantization benefits.

### E.3.2 SCALABLE DEBATE INFERENCE FRAMEWORK

The inference system enables efficient multi-agent debates through adapter hot-swapping and dynamic resource management, designed to scale seamlessly with available compute resources. A centralized base model manager maintains a pool of quantized base models shared across agents, implementing lazy loading and caching for lightweight LoRA adapters that enable rapid agent switching without full model reloading. An adapter job scheduler coordinates debate workloads across available GPUs through adaptive load balancing, priority queues for debate tasks ordered by round index, and automatic cluster adaptation that detects available hardware and distributes workloads accordingly. This architecture enables concurrent multi-agent inference on hardware that would typically support only single full-precision models, with debate throughput scaling linearly with compute resources while maintaining memory efficiency through quantization and adapter sharing.

### E.4 DEBATE PROMPT TEMPLATES

This appendix details the prompts used in our multi-agent debate system.

### E.4.1 INITIAL QUESTION PROMPT

Used in the first round of debate.

**For GPQA, MathQA, CommonsenseQA**

**Initial Prompt: Multiple Choice**

```
Answer the following multiple choice question as accurately
    as possible.
{question}

Provide a bullet point summary of your step-by-step reasoning
    .
Your final answer should be a single choice letter in the
    form
\boxed{answer}, at the end of your response.
```

**For GSM8K, SVAMP, AMC, MATH**

**Initial Prompt: Math Problem**

```
Solve the following math problem. {question}

Provide a bullet point summary of your step-by-step reasoning
    .
Your final answer should be a single numerical number, in the
    form
\boxed{answer}, at the end of your response.
```

### E.4.2 DEBATE PHASE PROMPT

Used in subsequent rounds of debate to incorporate peer context.

**Debate Phase Prompt**

```
Here are solutions from other agents:
{agent_responses}

Using each response as additional advice, give an updated
    bullet point summary of your step-by-step reasoning to
    the question: {question}

Make sure your final answer is in the form \boxed{answer}, at
    the end of your response.
```

## F ABLATION: GROUND TRUTH VS DEBATE MAJORITY VOTE

Tables 11 and 12 report ablations comparing post-training on oracle ground-truth (GT) labels versus (DMV) labels across MATH and GSM8K benchmarks.

In the single-agent setting (Table 11), DMV supervision proves consistently competitive with, and often superior to, GT supervision across Qwen2B, Llama-3B, Phi-4B, and Llama-8B. Gains are most pronounced for preference-based objectives such as KTO and DPO, where DMV provides a more stable learning signal. By contrast, the oracle GT signal sometimes produces degraded performance, particularly in KTO. Notably, DMV provides a robust alternative that avoids these pitfalls, often leading to stronger outcomes without requiring external supervision.

Table 11: Single-agent accuracy: Unsupervised post-training using debate-majority-vote–derived supervision (DMV) performs comparably to or outperforms supervised training using ground truth (GT) labels. Bold indicates the better score in each pair.

| Model | Dataset | Debate | SFT | | KTO | | DPO | | GRPO | |
|-------|---------|--------|-----|-----|-----|-----|-----|-----|------|-----|
| | | | *GT* | *DMV* | *GT* | *DMV* | *GT* | *DMV* | *GT* | *DMV* |
| Qwen2B | MATH | 7.67 | 12.13 | **12.20** | 12.67 | **19.67** | 17.13 | **23.00** | **21.33** | 17.33 |
| | GSM8K | 23.00 | 24.20 | **24.80** | 45.40 | **47.20** | **46.47** | 44.20 | **39.73** | 32.07 |
| Llama3B | MATH | 27.87 | **26.80** | 25.60 | 12.53 | **39.20** | 38.40 | **40.67** | **36.80** | 35.00 |
| | GSM8K | 57.33 | 54.13 | **56.73** | 27.27 | **66.27** | 56.87 | **66.73** | **54.93** | 52.20 |
| Phi4B | MATH | 34.60 | 33.07 | **35.20** | **33.73** | 33.00 | **37.00** | 36.33 | **37.80** | 37.33 |
| | GSM8K | 67.27 | **71.67** | 70.20 | **76.13** | 75.47 | 75.80 | **76.60** | **74.80** | 70.40 |
| Llama8B | MATH | 22.93 | **23.73** | 23.00 | **41.20** | 39.40 | 45.13 | **46.40** | 29.07 | **31.13** |
| | GSM8K | 57.93 | 40.20 | **41.67** | **72.60** | 70.87 | 76.33 | **77.67** | 61.27 | **66.87** |

Table 12: Multi-agent accuracy: Unsupervised post-training using debate-majority-vote–derived supervision (*DMV*) performs comparably to or outperforms supervised training using ground truth (*GT*) labels. Bold indicates the better score in each pair.

| Model | Dataset | Init | SFT | | KTO | | DPO | | GRPO | |
|-------|---------|------|-----|-----|-----|-----|-----|-----|------|-----|
| | | | *GT* | *DMV* | *GT* | *DMV* | *GT* | *DMV* | *GT* | *DMV* |
| Qwen2B | MATH | 32.4 | 38.6 | **40.6** | 40.4 | **46.2** | 39.4 | **41.2** | **43.6** | 40.2 |
| | GSM8K | 49.6 | 50.0 | **51.6** | **66.2** | 63.8 | **61.6** | 57.0 | **60.4** | 51.8 |
| Llama3B | MATH | 37.8 | **35.6** | 33.6 | 15.8 | **51.8** | 50.8 | **51.4** | 49.2 | **50.8** |
| | GSM8K | 65.6 | **65.8** | 64.0 | 29.4 | **73.8** | 67.0 | **73.8** | 64.4 | **69.6** |
| Phi4B | MATH | 44.4 | 44.0 | **46.0** | **46.8** | 44.6 | **50.2** | 48.6 | **51.4** | 50.8 |
| | GSM8K | 79.6 | **79.8** | 78.4 | **82.6** | 81.2 | 83.8 | **84.4** | 81.6 | **83.4** |
| Llama8B | MATH | 32.8 | **34.0** | 33.4 | **56.0** | 55.8 | **61.8** | 60.8 | **48.6** | 44.8 |
| | GSM8K | 74.0 | **65.2** | 65.2 | **81.2** | 79.4 | 81.6 | **83.0** | 83.2 | **84.2** |

In the multi-agent setting (Table 12), the advantages of DMV supervision become even clearer. Across nearly all models and methods, DMV either matches or exceeds GT labels. The gains are especially consistent under preference-learning formulations (KTO, DPO), where DMV supervision yields more reliable improvements to consensus-based performance. While GT labels retain competitive strength in certain cases (e.g., Phi-4B on MATH tasks), DMV repeatedly delivers higher or more stable final-round accuracies.

Together, these results show that debate-derived majority-vote supervision provides an effective, scalable alternative to oracle ground truth. DMV not only mitigates the instability observed when GT is used in preference-learning objectives, but also enhances both single- and multi-agent training. These findings underscore the efficacy of unsupervised alignment signals, leveraging a model's own consensus dynamics, as a robust substitute for human-labeled supervision.

# G    ABLATION: EFFECT OF PEER CONTEXT IN MULTI-AGENT DEBATE

Tables 13 and 14 examine whether conditioning on peer responses during training improves debate performance.

Training with peer context substantially improves multi-agent debate performance (Table 14). While initial round (single-agent) performance shows modest benefits from context training (5 out of 8 cases improve), the advantages become pronounced in final-round multi-agent settings. Context training improves final-round individual accuracy in 7 out of 8 cases, with particularly large gains for models like L8B-MATH (57.80 vs. 51.47) and L3B-MATH (45.80 vs. 41.13). Most importantly, final-round majority voting benefits from context in 6 out of 8 cases, with GSM8K tasks show-

Table 13: Multi-agent comparison of Context vs No Context runs across Initial Debate, SFT, KTO, DPO, and GRPO. Bold indicates which setting performed better in each pair.

| Model | Dataset | Debate | SFT | | KTO | | DPO | | GRPO | |
|---|---|---|---|---|---|---|---|---|---|---|
| | | | *Context* | *NoCtx* | *Context* | *NoCtx* | *Context* | *NoCtx* | *Context* | *NoCtx* |
| Qwen2B | MATH | 32.4 | **40.6** | 38.0 | 46.2 | **46.6** | 41.2 | **43.4** | **40.2** | 39.8 |
| | GSM8K | 49.6 | **51.6** | **51.6** | **63.8** | 59.2 | 57.0 | **57.8** | 51.8 | **57.2** |
| Llama3B | MATH | 37.8 | 33.6 | **37.4** | **51.8** | 27.0 | **51.4** | 51.0 | **50.8** | 47.0 |
| | GSM8K | 65.6 | 64.0 | **65.4** | **73.8** | 59.0 | **73.8** | 70.6 | **69.6** | 50.0 |
| Phi4B | MATH | 44.4 | **46.0** | 45.0 | 44.6 | **46.0** | **48.6** | 48.4 | 50.8 | **52.8** |
| | GSM8K | 79.6 | 78.4 | **80.0** | 81.2 | **82.0** | **84.4** | 81.8 | 83.4 | **82.0** |
| Llama8B | MATH | 32.8 | **33.4** | 31.4 | **55.8** | 49.2 | **60.8** | 57.2 | **44.8** | 44.6 |
| | GSM8K | 74.0 | 65.2 | **67.4** | 79.4 | **80.6** | **83.0** | 81.6 | **84.2** | 71.0 |

Table 14: Post-training with peer context teaches agents to utilize other agents' responses in the debate format for more effective final round ensemble reasoning (MV-DPO).

| | Initial Round Avg | | Final Round Avg | | Final Round MV | |
|---|---|---|---|---|---|---|
| Model-Data | NoCtx | Context | NoCtx | Context | NoCtx | Context |
| Qwen2B-MATH | 22.33 | **23.00** | 37.13 | **38.33** | **43.40** | 41.20 |
| Qwen2B-GSM8K | **46.53** | 44.20 | 52.53 | **54.73** | **57.80** | 57.00 |
| Llama3B-MATH | 40.27 | **40.67** | 41.13 | **45.80** | 51.00 | **51.40** |
| Llama3B-GSM8K | 65.87 | **66.73** | 67.87 | **70.60** | 70.60 | **73.80** |
| Phi4B-MATH | **39.47** | 36.33 | **39.53** | 34.73 | 48.40 | **48.60** |
| Phi4B-GSM8K | 74.67 | **76.60** | 74.60 | **77.80** | 81.80 | **84.40** |
| Llama8B-MATH | **49.87** | 46.40 | 51.47 | **57.80** | 57.20 | **60.80** |
| Llama8B-GSM8K | 75.80 | **77.67** | 78.00 | **81.20** | 81.60 | **83.00** |

ing consistent improvements across all model sizes (1.4-3.2 percentage points) and larger models achieving substantial gains on MATH tasks (e.g., L8B-MATH improving from 57.20 to 60.80). These results demonstrate that context-aware training teaches agents to effectively leverage peer feedback during deliberation, leading to stronger consensus outcomes in multi-agent debate.

# H  ITERATIVE IMPROVEMENT

Table 15 examines whether iterative training beyond the first iteration yields continued improvements. While It-1 produces the substantial gains reported in our main results, iterations 2 and 3 demonstrate continued modest improvements: in 23 of 24 evaluation settings, either It-2 or It-3 achieves the best performance. For example, Phi-4B on MATH majority vote increases from 55.00 (It-1) to 57.40 (It-3), and Llama-8B on GSM8K improves from 82.80 to 85.60.

These gains show clear diminishing returns compared to the It-0→It-1 jump, with typical improvements of 1-3 percentage points between iterations. Performance occasionally dips between adjacent iterations (e.g., L3B-GSM8K majority vote: 72.00→74.60→70.80) but generally trends upward. This pattern suggests that iterative training continues to extract useful signal from debate-generated data, though with decreasing marginal benefit after the initial iteration.

# I  DPO PAIR SELECTION STRATEGY ANALYSIS

We compared two strategies for creating preference pairs from multi-agent debate data: majority/minority partitioning versus confidence-based selection using model log-probabilities. Analysis covered 2,226 agent responses from 742 problems across multiple models and datasets. Majority/minority partitioning substantially outperforms confidence-based selection across all metrics. Majority responses achieve 68.0% accuracy versus 28.1% for minority responses (39.8 percentage point gap), while high-confidence responses achieve only 51.5% accuracy versus 33.2% for low-

Table 15: Iterative alternation between debate generation and post-training across four iterations shows substantial initial gains (It-0 to It-1) followed by diminishing returns. Model abbreviations: Q2B=Qwen-2B, L3B=Llama-3B, L8B=Llama-8B, P4B=Phi-4B.

| Model-Data | Initial Round Avg | | | | Final Round Avg | | | | Final Round MV | | | |
|---|---|---|---|---|---|---|---|---|---|---|---|---|
| | It-0 | It-1 | It-2 | It-3 | It-0 | It-1 | It-2 | It-3 | It-0 | It-1 | It-2 | It-3 |
| Q2B-MATH | 7.67 | 17.40 | 18.00 | **19.33** | 21.47 | 43.13 | 40.73 | **43.93** | 32.40 | 47.40 | 41.20 | **48.60** |
| Q2B-GSM8K | 23.00 | 44.20 | 44.67 | **44.73** | 35.07 | 58.67 | **59.93** | 59.47 | 49.60 | 60.80 | 62.20 | **62.60** |
| L3B-MATH | 27.87 | 40.93 | 39.33 | **41.93** | 24.27 | 48.27 | 46.93 | **48.73** | 37.80 | **55.00** | 54.00 | 53.60 |
| L3B-GSM8K | 57.33 | 64.60 | **67.07** | 65.20 | 49.20 | 68.73 | **71.13** | 69.07 | 65.60 | 72.00 | **74.60** | 70.80 |
| P4B-MATH | 34.60 | 43.27 | 43.67 | **43.80** | 34.37 | 48.00 | 48.93 | **50.20** | 44.40 | 55.00 | 55.80 | **57.40** |
| P4B-GSM8K | 67.27 | 75.73 | **76.20** | 75.47 | 68.53 | 77.00 | **79.20** | 77.60 | 79.60 | 81.40 | **84.40** | 83.00 |
| L8B-MATH | 22.93 | 44.53 | 44.67 | **44.87** | 22.53 | 55.73 | **57.60** | 56.07 | 32.80 | 58.20 | **60.40** | 59.80 |
| L8B-GSM8K | 57.93 | 77.00 | 77.87 | **78.80** | 56.53 | 80.80 | 80.73 | **82.93** | 74.00 | 82.80 | 82.60 | **85.60** |

confidence (18.3 percentage point gap). The majority strategy yields an effect size of Cohen's d = 1.832 compared to 0.281 for confidence-based selection, a 6.5× difference in discriminative power. Additionally, majority voting provides usable preference signals in 70.5% of examples versus 35.2% for confidence-based selection. These results validate using majority vote consensus for DPO pair selection, demonstrating that collective agreement provides more reliable quality signals than individual model confidence for mathematical reasoning tasks.

## J  SELF-CONSISTENCY AND ACCURACY CORRELATION ANALYSIS

We analyze the correlation between self-consistency and accuracy improvements across three experimental conditions to assess the robustness of our findings under different token generation limits and quantization settings. We test three conditions: (1) Token Capped + Quantized (256 tokens, 4-bit), our standard experimental setup matching computational constraints; (2) Token Uncapped + Non-Quantized (2048 tokens with no observed truncation, full precision), representing maximum generation quality without computational constraints; and (3) Token Uncapped + Quantized (2048 tokens with no observed truncation, 4-bit), a balanced approach removing token truncation while maintaining efficiency.

Table 16: Self-consistency vs accuracy correlation across experimental conditions. "Capped" refers to 256 token limit, which is used throughout the work due to computational constraints in multi-agent RL settings, and "Uncapped" refers to 2048 token limit where no response truncation occurred.

| Condition | Model-Dataset | Self-Consistency (%) | | Accuracy (%) | | Correlation |
|---|---|---|---|---|---|---|
| | | Base | Post | Base | Post | ($r$) |
| Capped +Quant | Llama3B-MathQA | 31.2 | **49.4** | 29.1 | **45.8** | 0.954 |
| | Qwen2B-Math | 12.0 | **27.8** | 11.6 | **26.8** | 0.989 |
| | Qwen2B-GSM8K | 26.4 | **54.0** | 25.9 | **52.7** | 0.982 |
| Uncapped +Full | Llama3B-MathQA | 49.5 | **68.5** | 45.1 | **60.9** | 0.869 |
| | Qwen2B-Math | 50.4 | **57.2** | 48.2 | **54.4** | 0.944 |
| | Qwen2B-GSM8K | 51.4 | **76.4** | 50.8 | **75.5** | 0.933 |
| Uncapped +Quant | Llama3B-MathQA | 49.8 | **67.3** | 45.8 | **60.8** | 0.899 |
| | Qwen2B-Math | 42.8 | **49.7** | 38.9 | **45.2** | 0.935 |
| | Qwen2B-GSM8K | 50.6 | **73.5** | 48.9 | **71.3** | 0.927 |

All conditions show strong positive correlations ($r > 0.86$) between self-consistency and accuracy, validating self-consistency as a robust proxy for model performance across computational settings. Interestingly, capped conditions show slightly higher correlations (mean $r = 0.975$) compared to uncapped conditions (mean $r = 0.915$ for both quantized and full precision). This tighter coupling likely reflects how post-training teaches more efficient reasoning: models learn to better utilize

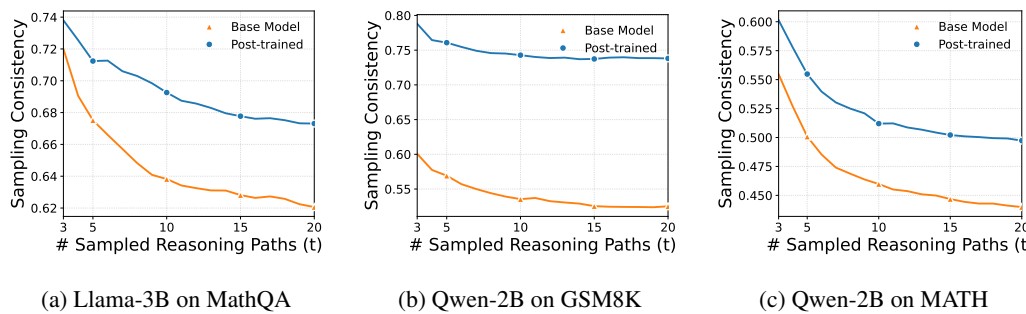

(a) Llama-3B on MathQA      (b) Qwen-2B on GSM8K      (c) Qwen-2B on MATH

Figure 5: Self-consistency improvements persist without token constraints. Models trained with 256-token debates still show gains when tested with full-length responses, though with reduced effect sizes due to the weaker training signal compared to testing conditions. Colors: Blue: post-trained model, Orange: base model.

limited token budgets, producing fewer truncated responses while achieving higher accuracy within constraints. In uncapped settings, post-trained models generate 22-36% shorter responses while still being more accurate, demonstrating that this efficiency persists without token limits. Four-bit quantization minimally impacts correlation strength, with quantized uncapped (mean $r = 0.920$) closely matching full precision uncapped (mean $r = 0.915$), demonstrating that computational efficiency can be achieved without degrading the consistency-accuracy relationship. Post-training consistently improves both metrics across all conditions, with self-consistency gains ranging from 6.9 to 27.6 percentage points and accuracy gains from 6.3 to 24.7 percentage points, confirming that our approach remains effective under varying computational constraints.

### J.1 SELF-CONSISTENCY IMPROVEMENTS WITHOUT TOKEN CONSTRAINTS

Figure 5 shows self-consistency curves for the aforementioned model-dataset pairs without token constraints. Improvements persist across all configurations, demonstrating that MACA's benefits are not merely artifacts of addressing truncation. The effect sizes are slightly smaller than in our main results due to train-test mismatch: these models were trained on debate signals from 256-token responses but tested without constraints. Training on full-length debates would likely yield larger improvements, as the consensus signals would be stronger and better aligned with test conditions.

## K IMPACT OF MACA ON GENERAL REASONING

We demonstrate that debate-derived majority vote supervision achieves comparable performance to ground truth supervision while enabling effective generalization to unseen reasoning domains. Tables 17 and 18 present direct comparisons between these supervision approaches across mathematical training domains and out-of-distribution tasks.

Table 17 shows that debate-derived supervision performs comparably to ground truth labels on training domains, with methods trading wins across model-dataset combinations. Table 18 reveals that both supervision approaches generalize effectively to unseen reasoning tasks, including mathematical word problems (SVAMP), science reasoning (GPQA), and commonsense reasoning (CSQA). Both methods show substantial improvements over base performance across all domains, confirming that MACA develops transferable reasoning capabilities.

These results demonstrate that debate-generated consensus signals provide an effective unsupervised alternative to ground truth supervision, achieving comparable performance without human annotation. This approach offers significant advantages for scaling reasoning improvements to new domains or large datasets where expert labels are unavailable or prohibitively expensive.

Table 17: Within-distribution performance comparison: Debate-derived majority vote supervision vs. ground truth supervision on training domains. Models are trained on MATH, GSM8K, and MathQA datasets. Bold indicates the better method for each model-dataset pair, demonstrating that debate-derived supervision achieves comparable performance to ground truth labels.

| Model | Method | MATH | | GSM8K | | MathQA | |
|---|---|---|---|---|---|---|---|
| | | Debate-MV | GT | Debate-MV | GT | Debate-MV | GT |
| Qwen2B | Base | 10.4 | 10.4 | 27.0 | 27.0 | 7.4 | 7.4 |
| | SFT | **10.8** | 10.4 | 25.6 | **26.4** | 8.2 | **8.8** |
| | GRPO | 19.4 | **21.0** | 45.2 | **48.6** | 18.6 | **19.6** |
| | KTO | 22.6 | **23.2** | **54.8** | 54.6 | **28.8** | 28.6 |
| | DPO | **24.8** | 24.2 | 51.4 | **52.0** | **24.2** | 24.0 |
| Llama3B | Base | 32.0 | 32.0 | 69.6 | 69.6 | 24.6 | 24.6 |
| | SFT | **33.2** | 32.4 | 64.2 | 64.2 | **26.4** | 25.2 |
| | GRPO | 45.8 | **46.4** | **75.8** | 74.8 | **36.2** | 31.8 |
| | KTO | **48.0** | 47.8 | 76.0 | **76.8** | **41.4** | 40.6 |
| | DPO | 53.2 | **53.6** | **80.4** | 77.8 | **46.2** | 45.4 |

Table 18: Cross-domain generalization: Debate-derived majority vote supervision vs. ground truth supervision on unseen reasoning tasks. Models trained on mathematical datasets (MATH, GSM8K, MathQA) generalize effectively to diverse reasoning domains. Bold indicates the better method for each model-dataset pair, showing that both supervision approaches transfer well to out-of-distribution tasks.

| Model | Method | SVAMP | | GPQA | | CSQA | |
|---|---|---|---|---|---|---|---|
| | | Debate-MV | GT | Debate-MV | GT | Debate-MV | GT |
| Qwen2B | Base | 48.30 | 48.30 | 0.45 | 0.45 | 3.80 | 3.80 |
| | SFT | **53.30** | 53.00 | **17.90** | 0.89 | 16.80 | **18.80** |
| | GRPO | **60.30** | 58.33 | **8.70** | 7.81 | **10.80** | 9.60 |
| | KTO | 76.00 | **76.33** | 16.70 | **16.96** | **63.40** | 60.80 |
| | DPO | **65.00** | 64.67 | 19.64 | **20.98** | **62.2** | 60.80 |
| Llama3B | Base | 71.30 | 71.30 | 0.67 | 0.67 | 53.00 | 53.00 |
| | SFT | 68.33 | **72.0** | **2.23** | 2.23 | 57.40 | **57.80** |
| | GRPO | 75.00 | **79.33** | **6.92** | 5.13 | **63.20** | 59.40 |
| | KTO | 76.70 | **78.67** | 8.93 | **9.82** | **62.20** | 61.80 |
| | DPO | 78.40 | **80.67** | **11.40** | 11.60 | **64.00** | 62.40 |

## L  POST-TRAINING METHOD IMPACT ON LOG-PROBABILITY DISTRIBUTION

Figure 6 shows how different post-training methods affected log-probability distributions for Qwen-2B on MATH. Preference-based methods (DPO, KTO) increased the density of majority distributions, with KTO showing particularly pronounced effects: higher peaks and tighter tails indicating more concentrated probability mass around consensus responses. SFT left the majority distribution largely unchanged while substantially reducing minority distribution density. GRPO created the most dramatic separation between distributions, shifting both leftward (lower log-probabilities overall) but with the minority distribution shifting much more substantially than the majority. While these different patterns (probability concentration for preference methods versus selective penalization for GRPO) all corresponded with performance improvements, further research is needed to better understand the relationship between these specific distributional changes and the impact on consistency.

## M  PREFERENCE LEARNING AS AN IMPLICIT FORMAT REWARD

This section investigates how preference learning through consensus signals acts as an implicit format reward, teaching models to produce more efficient and accurate reasoning without explicit formatting supervision. We examine three key aspects: the baseline capability requirements for ef-

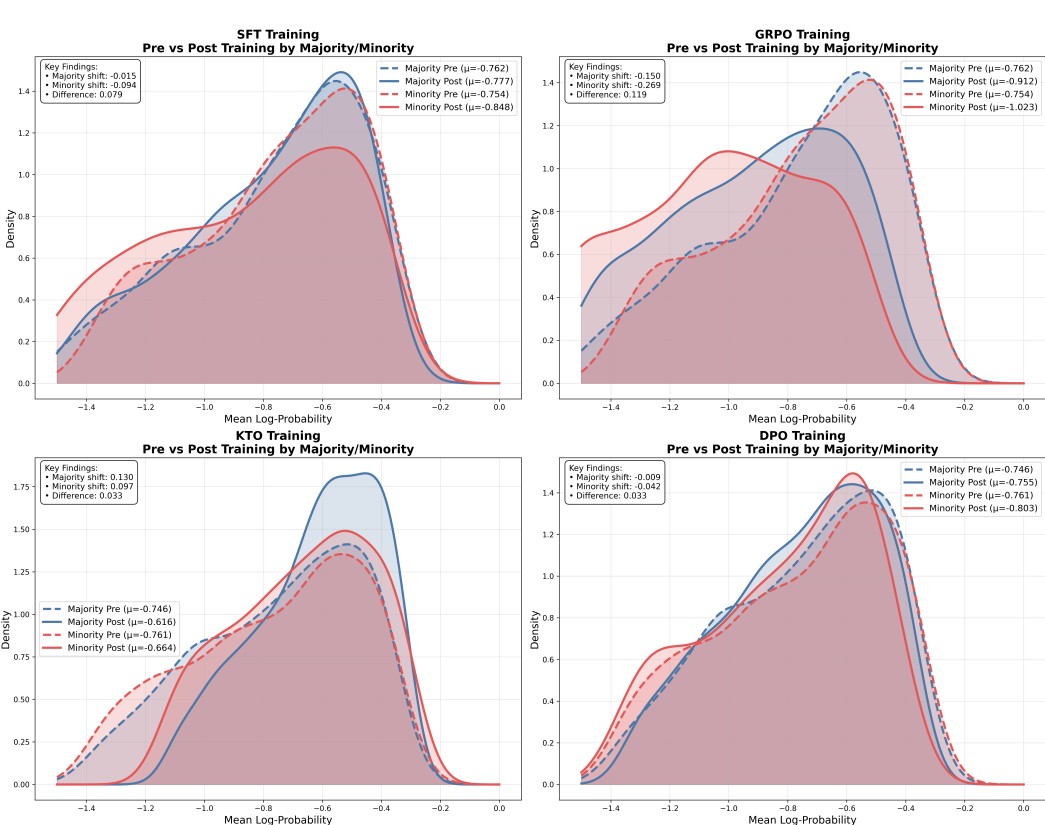

Figure 6: Log probability distributions for majority and minority answers before and after post-training (Qwen2B, Math).

fective consensus formation, how token constraints interact with reasoning improvements, and the decomposition of performance gains into format improvements versus problem-solving accuracy improvements. Crucially, we demonstrate that while preference learning does function effectively as an implicit format reward, most of the performance gains observed are attributed to fundamental improvements in problem-solving accuracy rather than mere formatting compliance.

## M.1 BASELINE CAPABILITY REQUIREMENTS FOR CONSENSUS FORMATION

Our experiments reveal that MACA requires sufficient baseline model capability to generate meaningful consensus signals. When models lack foundational problem-solving ability, they fail to produce the correct responses necessary for consensus-based reinforcement. Table 19 illustrates this limitation: under a strict 256-token limit, Qwen2B produces no correct responses on AMC, resulting in no useful consensus to reinforce. Increasing the token limit to 512 partially mitigates this by allowing more reasoning space, though inefficient reasoning patterns can still cause truncation.

Table 19: Impact of token limits and baseline capability on AMC performance. Shows accuracy percentages for base models and post-trained models ("All") under different token constraints. When models lack sufficient baseline capability (Qwen2B at 256 tokens), consensus formation fails as no correct responses are generated for reinforcement.

|  | max_new_tokens = 256 | | max_new_tokens = 512 | |
|  | Base | All | Base | All |
| --- | --- | --- | --- | --- |
| Qwen2B | 0.0 | 0.0 | 5.0 | **12.5** |
| Llama3B | 7.5 | **10.0** | 10.0 | **20.0** |

This baseline capability requirement has important implications for applying MACA: models must possess some initial problem-solving ability on the target domain to benefit from consensus-based training. However, once this threshold is met, we observe that improvements on easier datasets can generalize to more challenging tasks, suggesting that enhanced self-consistency helps overcome some limitations of consensus formation on difficult problems.

## M.2 TOKEN EFFICIENCY AND REASONING QUALITY IMPROVEMENTS

Post-training simultaneously addresses token efficiency and reasoning quality. Models trained with debate under constrained token limits generate more parseable answers within the budget while achieving higher accuracy on those answers. To verify that improvements extend beyond mere format optimization, we evaluated models post-trained with 256-token debates using 512-token test budgets.

Table 20 demonstrates that performance gains persist under increased token limits, indicating genuine reasoning enhancement rather than just format improvements. We also analyze the sequence lengths of the trajectories generated for the analysis in App. J, showing that uncapped post-trained models generate responses that are 22-36% shorter than their base models. This evidence suggests that self-guided preference learning functions as an implicit format reward, teaching models to produce more concise and effective reasoning patterns that generalize across computational budgets and task difficulties.

## M.3 DECOMPOSING THE SOURCES OF IMPROVEMENT

To understand whether our gains stem from improved reasoning or merely better formatting, we decompose performance improvements into their constituent components. Our analysis separates gains from better completion (avoiding truncation) versus fundamental reasoning improvements.

The results reveal that 69-100% of improvements stem from better reasoning rather than just avoiding truncation. While base models exhibit high truncation rates (e.g., 74.8% for Qwen2B on CSQA, 82.8% on MATH), post-training reduces these dramatically. However, even after accounting for completion improvements, substantial reasoning gains remain, confirming that our method teaches fundamentally better problem-solving, not merely more efficient token usage.

Table 20: Improvements from post-training with max_new_tokens = 256 translate when tested with larger token counts (512). "All" represents a model post-trained on Math, MathQA, and GSM8K; its row reports improvement deltas over the base model performance.

| Model | Trained on | Tested on (max_new_tokens = 512) | | | | | |
|-------|-----------|------|-------|--------|-------|-----|------|
|       |           | MATH | GSM8K | MathQA | SVAMP | AMC | GPQA |
| Qwen2B | Base | 32.8 | 47.4 | 18.8 | 53.7 | 7.5 | 12.5 |
|        | All  | ↑15.6 | ↑24.6 | ↑22.2 | ↑20.0 | ↑2.5 | ↑11.2 |
| Llama3B | Base | 60.8 | 76.8 | 43.0 | 78.0 | 10.0 | 9.2 |
|         | All  | ↑6.4 | ↓0.2 | ↑11.2 | ↑3.0 | ↑10.0 | ↑14.5 |

Table 21: Decomposition of performance improvements into completion gains versus reasoning gains. "From Completion" represents accuracy gained if all truncated base model responses had been allowed to complete. "From Better Reasoning" represents the remaining improvement attributable to fundamental problem-solving enhancement. Percentages show the relative contribution of each component to total gains.

| Model | Dataset | Before | After | Total Gain | From Completion | From Better Reasoning |
|-------|---------|--------|-------|------------|-----------------|----------------------|
| Qwen2B | CSQA | 3.8 | 63.4 | ↑59.6 | ↑18.6 (31%) | ↑41.0 (69%) |
|        | GPQA | 0.4 | 16.7 | ↑16.3 | ↑0.7 (4%) | ↑15.6 (96%) |
|        | GSM8K | 24.6 | 54.8 | ↑30.2 | ↑0.0 (0%) | ↑30.2 (100%) |
|        | MATH | 10.4 | 22.6 | ↑12.2 | ↑0.0 (0%) | ↑12.2 (100%) |
|        | MathQA | 7.4 | 28.8 | ↑21.4 | ↑2.0 (9%) | ↑19.4 (91%) |
|        | SVAMP | 48.3 | 76.0 | ↑27.7 | ↑0.4 (1%) | ↑27.3 (99%) |
| Llama3B | CSQA | 53.0 | 64.0 | ↑11.0 | ↑1.2 (11%) | ↑9.8 (89%) |
|         | GPQA | 0.7 | 11.4 | ↑10.7 | ↑0.9 (8%) | ↑9.8 (92%) |
|         | GSM8K | 69.6 | 80.4 | ↑10.8 | ↑0.2 (2%) | ↑10.6 (98%) |
|         | MATH | 41.2 | 53.2 | ↑12.0 | ↑0.2 (2%) | ↑11.8 (98%) |
|         | MathQA | 29.2 | 46.2 | ↑17.0 | ↓0.2 (-1%) | ↑17.2 (101%) |
|         | SVAMP | 71.3 | 75.7 | ↑4.3 | ↑0.0 (0%) | ↑4.3 (100%) |

## M.4 FORMATTING IMPROVEMENTS AND REMAINING REASONING GAINS

To further isolate reasoning improvements from formatting effects, we analyze how post-training affects answer formatting compliance. Table 22 shows the percentage of responses with correct reasoning but incorrect formatting, and calculates reasoning gains that persist beyond all formatting improvements. Post-training dramatically reduces formatting losses (e.g., from 38.8% to 0.6% on CSQA for Qwen2B). However, the "Reasoning Gain Beyond Formatting" column reveals that substantial improvements remain even after perfect formatting is assumed. For instance, Qwen2B's 59.6 percentage point improvement on CSQA includes 38.2 points from better formatting; the remaining 21.4 points represent accuracy improvements distinct from formatting.

## M.5 PARSER IMPLEMENTATION DETAILS

To quantify formatting improvements versus reasoning gains, we employed two parsers with different strictness levels. Our standard parser requires answers in \boxed{} format and returns None for any deviation. The relaxed parser, used for impact analysis in Table 22, accepts common natural language patterns like "The answer is A" or "Answer: 42" by searching the entire response for valid answer formats.

This relaxed parser first attempts the strict extraction, then falls back to regex patterns that capture answers expressed naturally in text. For multiple choice, it accepts patterns like "[Tt]he answer is ([A-E])" or "Answer: ([A-E])". For numerical answers, it extracts from patterns like "= 42" at line endings or "The final answer is 42". When multiple patterns match, it takes the last occurrence, mimicking how humans identify the final answer in a reasoning chain.

Table 22: Analysis of formatting improvements versus reasoning gains. "Accuracy Lost to Format Errors" shows the percentage of responses with correct reasoning but incorrect formatting (e.g., writing "The answer is A" instead of \boxed{A}). "Reasoning Gain Beyond Formatting" shows improvements that persist even after accounting for all formatting fixes, calculated as: Total Gain - (Format Loss Before - Format Loss After).

| Model | Dataset | Accuracy Lost to Format Errors (%) | | Reasoning Gain Beyond Formatting (%) |
| | | Before Training | After Training | |
|---|---|---|---|---|
| Qwen2B | CSQA | 38.8 | 0.6 | ↑21.4 |
| | GPQA | 6.2 | 1.6 | ↑11.7 |
| | MathQA | 5.8 | 0.6 | ↑16.2 |
| Llama3B | CSQA | 7.0 | 1.4 | ↑5.4 |
| | GPQA | 7.6 | 3.8 | ↑6.9 |
| | MathQA | 5.8 | 2.8 | ↑14.0 |

The gap between strict and relaxed parser accuracies precisely measures the "Accuracy Lost to Format Errors" in Table 22. Strict formatting compliance is a key measure of a model's ability to follow instructions and is critical for downstream applications that rely on reliable parsing of LM outputs, while the relaxed parser accounts for human-interpretable correctness. This dual evaluation reveals that preference learning through consensus not only teaches proper formatting as an implicit reward but fundamentally improves problem-solving capabilities, with reasoning gains persisting even when formatting constraints are removed.

## N    TRAINING CURVES

The training dynamics of our post-training methods provide insight into how models learn to refine responses towards multi-agent consensus preferences. Figures 7a–8b show example training curves across key metrics during post-training with MV-DPO, MV-KTO, MV-GRPO, and MV-SFT.

Across all methods, the reward margins between chosen (consensus) and rejected (non-consensus) responses increase consistently, indicating effective preference learning. MV-DPO and MV-KTO reveal this pattern strongly: margins start near zero and grow steadily as the models optimize towards favoring majority-preferred outputs. MV-KTO achieves similar reward improvements despite not requiring strict paired comparisons, demonstrating its robust learning dynamics.

Training accuracy converges for all methods, reaching high classification levels between consensus and non-consensus responses, showing that each approach successfully reinforces desired sampling behaviors. Correspondingly, losses decrease smoothly without signs of instability or collapse, indicating stable training processes.

Log probabilities of rejected responses decline across post-training methods, reflecting the models' increasing tendency to assign lower likelihood to outputs outside the consensus. This is most pronounced in MV-DPO and MV-GRPO, where rejected rewards fall more steeply, delineating a clear separation between preferred and discouraged responses. MV-SFT, operating via imitation learning, exhibits strong improvements in token accuracy and loss while reinforcing consensus-aligned responses effectively.

Overall, the asymmetric reward trajectories, where chosen response rewards remain relatively stable or increase slightly while rejected response rewards decline sharply, suggest that post-training primarily discourages generation of minority or outlier outputs. This mechanism is a core driver behind the improved sampling consistency and reasoning quality observed in our experiments.

These training curves collectively support our hypothesis that post-training with majority vote preferences enables models to internalize collective agreement notions and reproduce responses better aligned with multi-agent consensus.

Table 23: Single-agent: RL with debate outperforms RL with single-round majority vote. Benchmark against TTRL (single-round R0 MV-GRPO), ScPO (single-round R0 MV-DPO), MACA (debate final round R1 MV-DPO with debate context after collaborative exchange). Accuracy averaged over 3 seeds with standard deviation reported in comparison to base(instruction-tuned) model.

|  | Qwen-2B | Llama-3B | Phi-4B | Llama-8B |
|---|---|---|---|---|
| **MATH** | | | | |
| Base | 9.4 | 40.2 | 39.8 | 36.0 |
| R0 MV-GRPO | **27.4 ± 2.9** | 45.5 ± 5.7 | 45.9 ± 2.1 | 43.5 ± 0.2 |
| R0 MV-DPO | 11.7 ± 1.1 | 43.6 ± 0.2 | 39.9 ± 0.5 | 39.7 ± 0.6 |
| R1 MV-DPO | 26.1 ± 0.4 | **52.7 ± 0.7** | **46.7 ± 0.2** | **53.1 ± 0.8** |
| **GSM8K** | | | | |
| Base | 26.4 | 72.8 | 76.4 | 67.8 |
| R0 MV-GRPO | **50.9 ± 1.6** | 50.5 ± 8.3 | 81.2 ± 1.6 | 54.3 ± 5.7 |
| R0 MV-DPO | 42.3 ± 3.4 | 74.4 ± 0.8 | 82.3 ± 0.5 | 80.7 ± 1.2 |
| R1 MV-DPO | 50.2 ± 0.9 | **76.5 ± 0.3** | **83.7 ± 0.3** | **86.0 ± 0.2** |

## O  POST-TRAINING SELF-CONSISTENCY IMPROVEMENTS TRANSLATE FROM 4-BIT QUANTIZED MODEL TO FULL MODEL

As shown in Figures 9-11, post-training improvements on 4-bit quantized models transfer to full-precision models, and the improvement margin persists with larger numbers of trajectories sampled ($t = 50$).

## P  IMPACT OF POST-TRAINING ON DEBATE AGREEMENT RATES

Figure 12 shows how agent agreement patterns evolve through debate, comparing base models (Iteration 0) against consensus-trained models (Iteration 1). These distributions directly measure whether our training successfully shifts probability mass toward consensus trajectories $\mathcal{G}_+(x)$.

Base models show relatively uniform agreement distributions, with only 13.4% of examples achieving full consensus (3/3 agreement) in the final round. After consensus post-training with MV-DPO, this increases over three-fold to 43.4%, with systematic improvements across all agreement levels: increased mass at 2/3 and 3/3 agreement, decreased mass at 1/3 agreement. This redistribution confirms that training drives the policy toward consensus-supporting trajectories.

Additionally, unparseable responses drop from 11% to 0.6% without explicit format rewards. Since our training favors consensus completions, which must finish within token limits to be comparable, the method implicitly rewards efficient, complete reasoning patterns. This suggests consensus alignment naturally encourages concise and coherent reasoning as a prerequisite for measurable agreement.

## Q  PERFORMANCE COMPARISON OF RL ON DEBATE VS. SINGLE-ROUND MAJORITY VOTE RL IN SINGLE-AGENT SETTING

Table 23 reports the full results referenced in Table 6, including standard deviation over 3 seeds. We find that MACA with final round majority vote with debate context outperforms single-round majority-vote RL with DPO (akin to ScPO (Prasad et al., 2024)) in all 8 configurations tested, and RL with GRPO (akin to TTRL (Zuo et al., 2025)) in 6 out of 8 configurations tested, with the other 2 configs being within the standard deviation.

## R  EFFICIENCY

Here we detail the compute performance tradeoffs of MACA's MV-DPO with debate context against other single-agent benchmark methods. ScPO requires 0.075-0.58 GPU hours achieving -0.6%

to +13.6% gains, TTRL requires 2.2-7.7 GPU hours achieving -26.4% to +18% changes, while MACA (MV-DPO with debate) requires 0.73-1.68 GPU hours achieving consistent positive gains. Our efficiency plots demonstrate that MV-DPO on debate provides the best performance-efficiency tradeoff—higher performance than ScPO/TTRL with comparable computational cost to ScPO and significantly lower cost than TTRL.

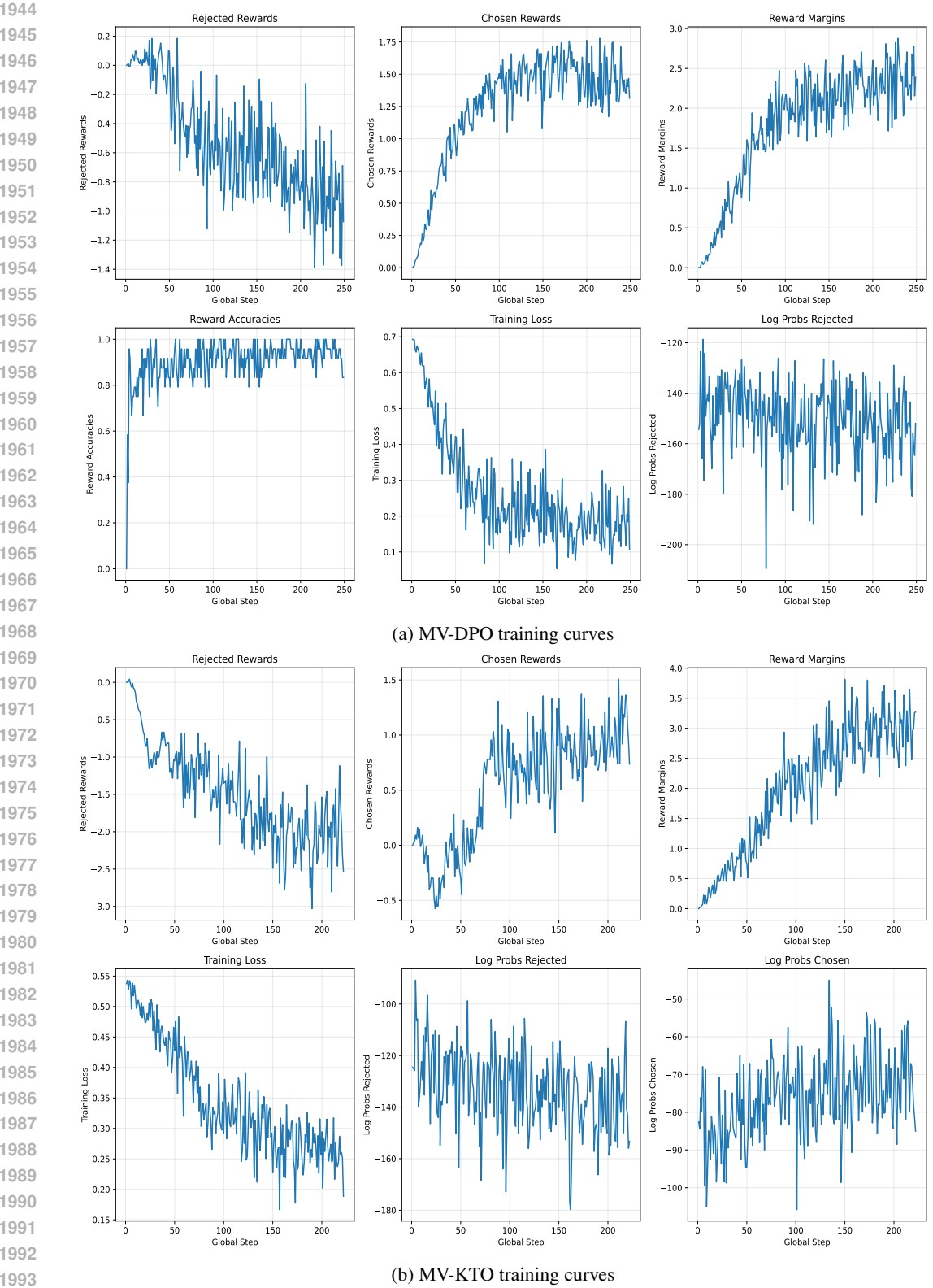

Figure 7: Training curves for MV-DPO and MV-KTO. (a) MV-DPO: increasing reward margin between majority (chosen) and minority (rejected) responses, with declining rejected log probabilities. (b) MV-KTO: similar reward margin increase using unpaired examples, with rejected log probabilities decreasing and chosen increasing.

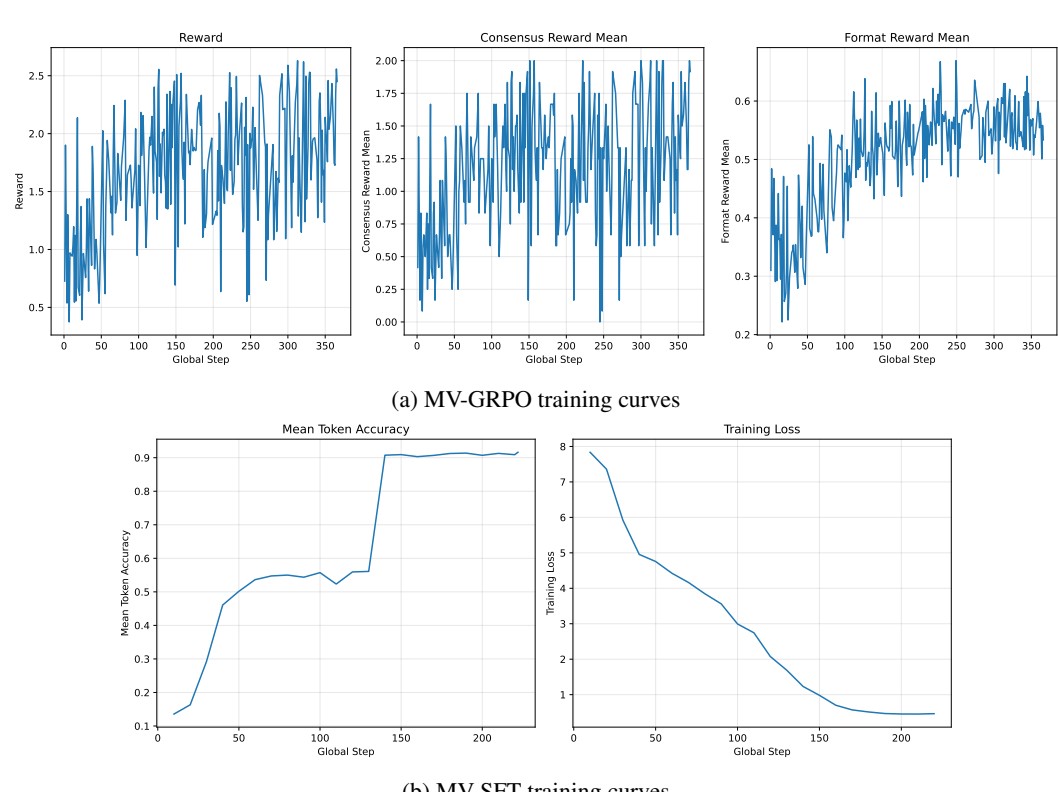

(a) MV-GRPO training curves

(b) MV-SFT training curves

Figure 8: Training curves for MV-GRPO and MV-SFT. (a) MV-GRPO: consensus and format rewards both increase. (b) MV-SFT: token accuracy increases while loss decreases.

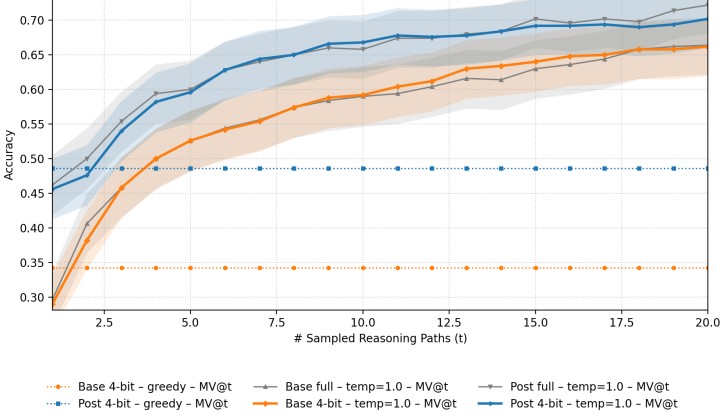

Figure 9: Llama-3B on MathQA (MV@t, t=20). Blue: post-trained 4-bit model, Orange: base 4-bit model, Grey: full-precision model.

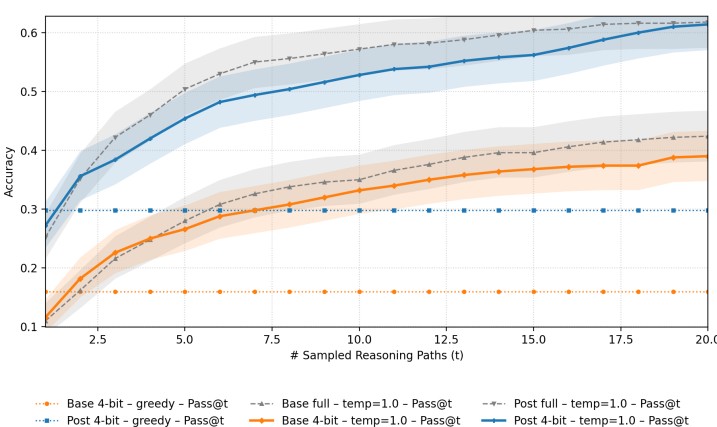

Figure 10: Qwen-2B on MATH (Pass@t, t=20). Blue: post-trained 4-bit model, Orange: base 4-bit model, Grey: full-precision model.

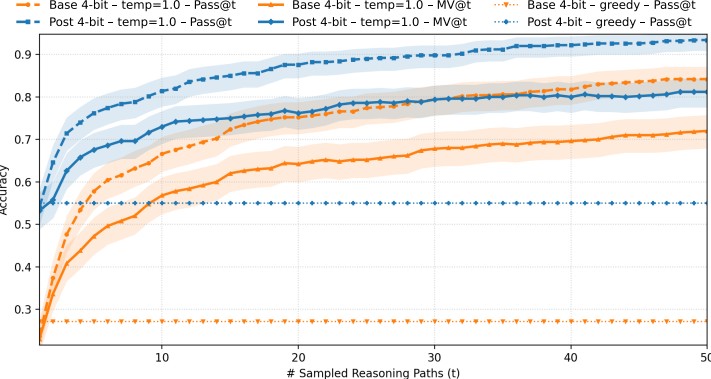

Figure 11: Qwen-2B on GSM8K (Pass@t, MV@t; t=50). Blue: post-trained 4-bit model, Orange: base 4-bit model.

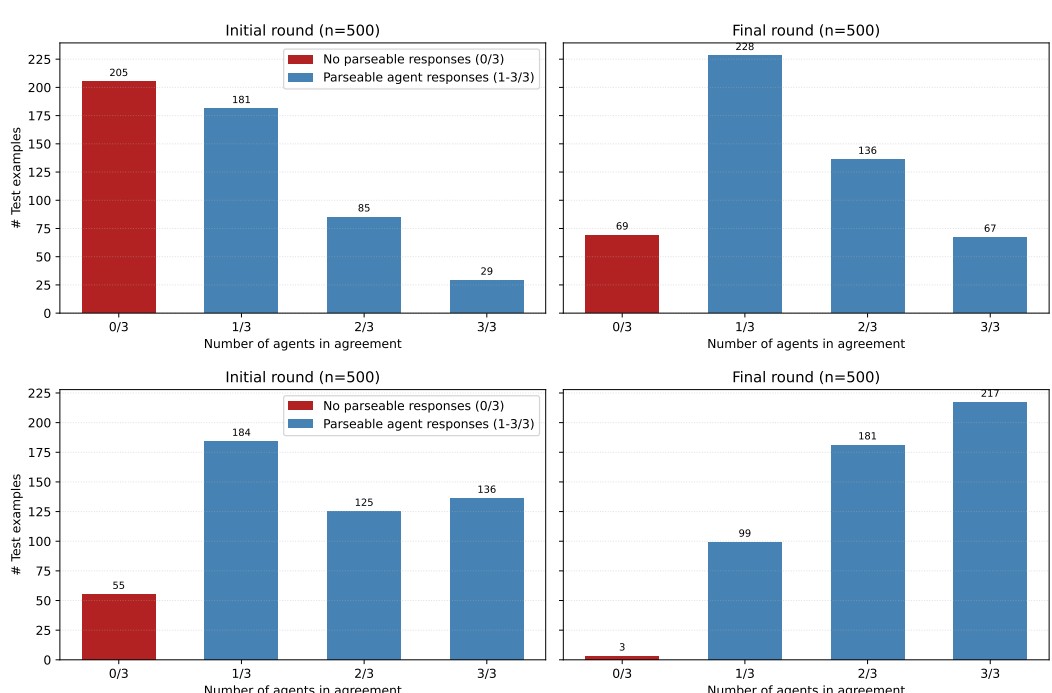

Figure 12: MACA drives meaningful improvements in both answer completeness and agent agreement, verifying the probability mass reallocation to the consensus set of reasoning trajectories (Ex: Qwen2B on GSM8K). Top: base model debate; bottom: post-trained model debate; left: initial debate round; right: final debate round.

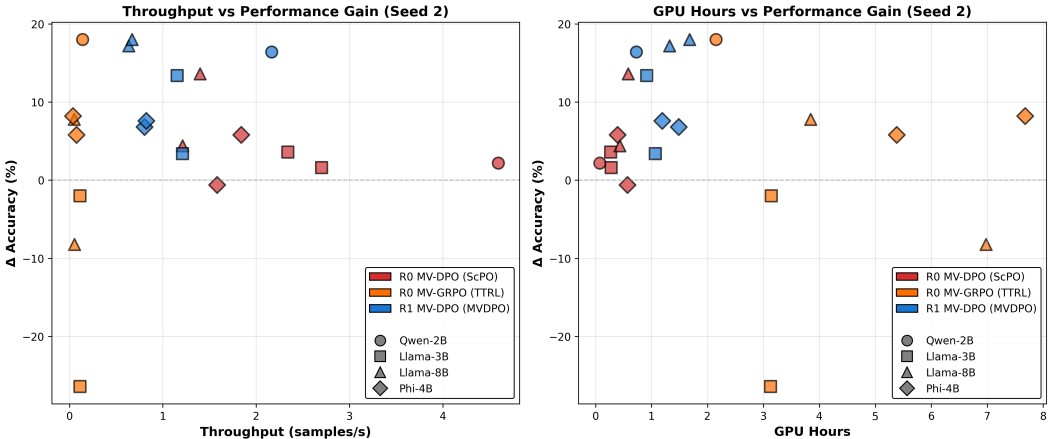

Figure 13: GPU throughput and training runtime compared with performance lift for single-agent post-training methods

