# OpenReview forum: "Internalizing Self-Consistency in Language Models: Multi-Agent Consensus Alignment"
_ICLR.cc/2026/Conference — ICLR 2026 Conference Desk Rejected Submission_

### Official Review · Reviewer_H9sy · 2025-10-25

**Soundness:** 2
**Presentation:** 2
**Contribution:** 1
**Rating:** 2
**Confidence:** 4

**Summary:**

In this work, the author argues that self-consistency should be a desired property of well-aligned reasoning models, and they address this by introducing MACA, a reinforcement learning framework that post-trains models to prefer consensus-supporting trajectories using majority/minority outcomes from multi-agent debate.

**Strengths:**

The proposed MACA framework can relax the need for ground truth labels and uses majority outcomes as the learning signal.

**Weaknesses:**

1. Learning from multi-agent debate trajectories is explored in prior work [1], as well as using the majority answer as the learning signal [2, 3]. In [1], the author also shows different levels of learning from the consensus-supporting and dissenting trajectories. This limits the novelty of this work.
2. Post-training methods like GRPO naturally sharpen the distribution and improve pass@1 performance substantially. This is similar to the motivation of this work about internalizing consistency and has been proven to be very effective. The author should compare with ScPO [2] and TTRL [3] to show how many improvements are coming from multi-agent debate, or the proposed method even outperforms these baselines.
3. The training is conducted on the base model instead of the instruction-tuned version. Comparing with instruction-tuned models is necessary, and it will be more convincing if the model is trained from instruction-tuned checkpoints.
4. Limiting responses to 256 tokens does not make sense to me. This budget is not sufficient for tasks like MATH and GPQA. Although in the appendix, this budget increases to 512, it is still too small.
5. The fact that "Unsupervised majority-vote signal is comparable to ground-truth" as shown in the ablation study is established in [2, 3], again limiting the novelty of this work.
6. The improvements are overclaiming since it is comparing the trained performance with the base model. The improvements should be compared with external baselines such as [2] and [3].
7. On the note of "how many improvements are coming from multi-agent debate", it is also important to compare with [1].


[1] https://arxiv.org/abs/2402.01620
[2] https://arxiv.org/abs/2411.04109
[3] https://arxiv.org/abs/2504.16084

**Questions:**

Please see weaknesses.

---

> ### Author Response · Authors · 2025-11-21
> **Reply to Reviewer H9sy (Part 1)**
>
> We sincerely thank you for your thorough feedback. We have addressed each of the weaknesses as listed below:
>
> ## W1, W5: Novelty vs. Prior Work [1,2,3]
>
> **Reviewer Concern:**
> > [In [1]] author also shows different levels of learning from the consensus-supporting and dissenting trajectories. This limits the novelty of this work.
>
> MADGI [1] uses teacher-generated debates with SFT for distillation, while MACA enables self-improvement through RL on the model's own debate signals—no teachers required. Tables 1-2 show RL consistently outperforms SFT (MV-DPO/KTO/GRPO vs MV-SFT). We also tested MACA against SFT-based debate distillation (W7 response), where RL outperforms SFT even with teacher signals, though we consider teacher-distillation beyond our self-improvement scope. The key distinction: MACA shows small models (2-8B params) can self-improve through their own multi-agent debates with RL, fundamentally different from MADGI's teacher distillation through SFT supervision paradigm.
>
> **Reviewer Concerns:**
> > Learning from multi-agent debate trajectories is explored in prior work [1], as well as using the majority answer as the learning signal [2, 3].
>
> > The fact that 'Unsupervised majority-vote signal is comparable to ground-truth' as shown in the ablation study is established in [2, 3], again limiting the novelty of this work.
>
> Our Related Work (p.2, lines 108-112) distinguishes MACA from ScPO [2] and TTRL [3], which use single-round majority vote with DPO and GRPO respectively. MACA instead "leverages multi-agent debate to produce higher-quality consensus signals from reasoning demonstrating peer grounding."
>
> We acknowledge our comparison should have been more explicit and have updated: (1) expanded Related Work, (2) new Methods paragraph on how MACA generalizes these approaches, and (3) revised Table 6 to include this comparison (please see W2 response).
>
> While [2, 3] showed single-round majority vote approximates ground truth, our contribution demonstrates that consensus from multi-agent debate with context—capturing how agents build upon each other's reasoning—provides stronger training signals. We validate this across multiple dimensions: debate vs. single-round, with/without context, preference learning vs. scalar rewards, and diverse models/datasets (Tables 4, 11-12, 17-18).
>
> Additionally, our finding that preference learning (DPO/KTO) outperforms scalar reward RL (GRPO) for debate consensus represents an empirical contribution not explored in prior works, as noted by Reviewer @d4Qb.
>
> ## W2: Performance Comparison Against Prior Work
>
> **Reviewer Concern:**
> > The author should compare with ScPO [2] and TTRL [3] to show how many improvements are coming from multi-agent debate, or the proposed method even outperforms these baselines.
>
> > The improvements should be compared with external baselines such as [2] and [3].
>
> We acknowledge that our comparison to ScPO and TTRL should have been more explicit—thank you for highlighting this. We have updated Table 6 to clearly label these comparisons and added corresponding discussion in the manuscript.
>
> **Direct comparison results (Table 6):**
> - **MACA vs ScPO:** MACA outperforms ScPO (single-round majority vote DPO) in all 8 settings tested
> - **MACA vs TTRL:** MACA outperforms TTRL (single-round majority vote GRPO) in 6 of 8 settings, within SD in other 2 settings
>
> *Arrows show gains over instruction-tuned model. Statistics gathered over 3 seeds.*
>
> | Method | **MATH** |||| **GSM8K** ||||
> |--------|----------|----------|---------|----------|----------|----------|---------|----------|
> | | Qwen-2B | Llama-3B | Phi-4B | Llama-8B | Qwen-2B | Llama-3B | Phi-4B | Llama-8B |
> | TTRL | ↑**18.0**±2.9 | ↑5.3±5.7 | ↑6.1±2.1 | ↑7.5±0.2 | ↑**24.5**±1.6 | ↓22.3±8.3 | ↑4.8±1.6 | ↓13.5±5.7 |
> | ScPO | ↑2.3±1.1 | ↑3.4±0.2 | ↑0.1±0.5 | ↑3.7±0.6 | ↑15.9±3.4 | ↑1.6±0.8 | ↑5.9±0.5 | ↑12.9±1.2 |
> | MV-DPO | ↑16.7±0.4 | ↑**12.5**±0.7 | ↑**6.9**±0.2 | ↑**17.1**±0.8 | ↑23.8±0.9 | ↑**3.7**±0.3 | ↑**7.3**±0.3 | ↑**18.2**±0.2 |
>
> **Note:** ↑ indicates improvement over instruction-tuned model. Bold values indicate best performance within each column.
>
> These results demonstrate that using consensus signals from multi-agent debate with context provides consistent improvements over single-round majority vote approaches. Our ablation studies (Tables 5, 13-14) provide additional detailed analysis showing how debate context contributes to the overall performance gains.

---

> > ### Author Response · Authors · 2025-11-21
> > **Reply to Reviewer H9sy (Part 2)**
> >
> > ## W3, W6: Clarification on Model Types
> >
> > **Reviewer Concerns:**
> > > The training is conducted on the base model instead of the instruction-tuned version. Comparing with instruction-tuned models is necessary.
> >
> > > The improvements are overclaiming since it is comparing the trained performance with the base model.
> >
> > We believe there might have been a misunderstanding regarding this point. We used instruction-tuned models throughout: Qwen2.5-1.5B-Instruct, Llama-3.2-3B-Instruct, Phi-3.5-mini-Instruct, and Llama-3.1-8B-Instruct (verifiable in our code). All reported improvements are relative to these instruction-tuned baselines, not base models. We've updated our Results section to make this explicit. Thank you for highlighting this clarity issue.
> >
> > ## W4: Token Limit Constraints
> >
> > **Reviewer Concern:**
> > > Limiting responses to 256 tokens does not make sense to me. This budget is not sufficient for tasks like MATH and GPQA.
> >
> > We imposed token limits during training due to computational constraints inherent in multi-agent RL with LLMs. However, extensive validation shows this doesn't diminish our findings: Table 16 demonstrates performance improvements persist when models are evaluated without token caps and without quantization, and results with 512-token limits (pp. 31-32) show similar gains. Since performance improvements remain robust regardless of evaluation constraints, training without token limits would likely yield even stronger results from richer debate signals. The key point is that our core finding—consensus signals from multi-agent debate with context providing stronger training signals than single-round majority vote—remains valid regardless of token constraints, as relative improvement matters for validating our method, not absolute performance under computational limitations.
> >
> > ## W7: Comparison with [1] for Debate Improvements
> >
> > **Reviewer Concern:**
> > > On the note of 'how many improvements are coming from multi-agent debate', it is also important to compare with [1].
> >
> > As we established in our response to W1, we do not feel MADGI is the relevant performance comparison in the context of reinforcement of self-generated training signals as it relies on SFT-distillation from debates between frontier models. Nevertheless, we ran an experiment using Llama-8B as a teacher model with Llama-3B (both instruction-tuned models), comparing MADGI's SFT-distillation from debate with MACA as an RL-distillation technique.
> >
> > | Training Signal Source | Model | MATH | GSM8K | Method |
> > |---|---|---|---|---|
> > | None | Llama-3B | 32.0 | 69.6 | Instruction-tuning |
> > | Llama-3B (self) | Llama-3B | 51.4 | 76.4 | MV-DPO |
> > | Llama-8B (teacher) | Llama-3B | 41.0 | 71.4 | MV-SFT |
> > | Llama-8B (teacher) | Llama-3B | 47.0 | 77.8 | MV-DPO |
> >
> > Results confirm that: (1) MV-DPO outperforms MV-SFT even with teacher signals (47.0 vs 41.0 on MATH), validating RL's superiority for debate-based learning, and (2) self-generated signals achieve competitive performance (51.4) without requiring a teacher model. While the relative benefits of self versus teacher debate signals merit further investigation, we focus on self-improvement without external supervision.
> >
> > Thank you for taking the time to review our paper. We believe that the clarifications and updates to the manuscript help better highlight the thoroughness and impact of our work. We would sincerely appreciate it if you would reconsider your score accordingly.
> >
> > ---
> >
> > **References:**
> >
> > [1] MADGI: Structured Distillation of Multi-Agent Interaction Graphs - https://arxiv.org/pdf/2402.01620
> > [2] ScPO - https://arxiv.org/pdf/2411.04109
> > [3] TTRL - https://arxiv.org/pdf/2504.16084

---

### Official Review · Reviewer_HLw1 · 2025-10-30

**Soundness:** 3
**Presentation:** 3
**Contribution:** 2
**Rating:** 4
**Confidence:** 3

**Summary:**

This paper introduces MACA, a self-supervised, reinforcement learning framework for post-training LLMs to enhance self-consistency in reasoning. The MACA approach formalizes self-consistency as an intrinsic property of LLMs and utilizes multi-agent debate, where multiple clones of an LLM generate, critique, and revise reasoning trajectories. Th framework generates training signals from these deliberative exchanges and optimizes LLMs using several objectives to internalize consensus rather than simple aggregation. Substantial empirical improvements are demonstrated over baseline and strong post-training baselines on mathematical, science, and commonsense reasoning benchmarks, with gains in both accuracy and self-consistency, generalization to unseen tasks, and efficiency in inference.

**Strengths:**

1. The MACA framework is not limited to GRPO; it can also be integrated with multi-agent RL and preference-based alignment objectives, suggesting broader applicability to LLM self-alignment settings.

2. The improvements in self-consistency observed on mathematical reasoning tasks transfer to science and commonsense benchmarks, indicating that the method generalizes beyond a single domain.

**Weaknesses:**

1. The core learning loop is closely aligned with recent test-time reinforcement learning approaches [1], in which multiple sampled reasoning trajectories are compared and the consensus outcome is used as a self-supervised preference signal to update the model. The main distinction in this paper is that the consensus signal is generated via multi-agent debate rather than independent sampling. However, this conceptual similarity should be made more explicit, and a direct comparison to test-time RL methods is necessary to clarify what is genuinely novel in the proposed contribution.

2. While the improvements over SFT baselines and previous post-training paradigms are clear, the paper would benefit from direct, quantitative comparison to more diverse non-MACA multi-agent aggregation schemes. It is unclear how much gain stems from the debate protocol itself versus just using more samples at training.

[1] Zuo et al., "TTRL: Test-time Reinforcement learning." 2025

**Questions:**

See weaknesses

---

> ### Author Response · Authors · 2025-11-19
> **Reply to Reviewer HLw1**
>
> Thank you for your thoughtful review and for recognizing that MACA's broad applicability to multiple RL objectives and strong cross-domain generalization are valuable contributions. We address your concerns below.
>
> ## W1: Comparison to Test-Time RL (TTRL)
>
> **Reviewer Concern:**
> > This conceptual similarity should be made more explicit, and a direct comparison to test-time RL methods is necessary to clarify what is genuinely novel.
>
> We acknowledge that our comparison to TTRL should have been more explicit. We have updated our manuscript to clarify this relationship:
>
> Our Related Work (p.2, lines 108-112) now states:
> > Majority vote RL methods such as TTRL (Zuo et al., 2025) and ScPO (Prasad et al., 2024) use GRPO and DPO, respectively, to reinforce single-round majority vote, whereas our framework leverages multi-agent debate to produce higher-quality consensus signals from reasoning demonstrating peer grounding...
>
> Updated Table 6 now explicitly compares MACA against these baselines:
> - **MACA vs ScPO [1]:** MACA outperforms ScPO (single-round majority vote DPO) in all 8 settings tested
> - **MACA vs TTRL [2]:** MACA outperforms TTRL (single-round majority vote GRPO) in 6 of 8 settings, within SD in other 2 settings
>
> *Arrows show gains over instruction-tuned model. Statistics gathered over 3 seeds.*
>
> | Method | **MATH** |||| **GSM8K** ||||
> |--------|----------|----------|---------|----------|----------|----------|---------|----------|
> | | Qwen-2B | Llama-3B | Phi-4B | Llama-8B | Qwen-2B | Llama-3B | Phi-4B | Llama-8B |
> | TTRL | ↑**18.0**±2.9 | ↑5.3±5.7 | ↑6.1±2.1 | ↑7.5±0.2 | ↑**24.5**±1.6 | ↓22.3±8.3 | ↑4.8±1.6 | ↓13.5±5.7 |
> | ScPO | ↑2.3±1.1 | ↑3.4±0.2 | ↑0.1±0.5 | ↑3.7±0.6 | ↑15.9±3.4 | ↑1.6±0.8 | ↑5.9±0.5 | ↑12.9±1.2 |
> | MV-DPO | ↑16.7±0.4 | ↑**12.5**±0.7 | ↑**6.9**±0.2 | ↑**17.1**±0.8 | ↑23.8±0.9 | ↑**3.7**±0.3 | ↑**7.3**±0.3 | ↑**18.2**±0.2 |
>
> **Note:** ↑ indicates improvement over instruction-tuned model. Bold values indicate best performance within each column.
>
> These results show that using consensus from deliberative agent discussion—with evidence of peer grounding in the debate context—is associated with consistent improvements over single-round majority vote. We thank you for highlighting the need to make these comparisons clearer.
>
> ## W2: Debate Protocol vs. More Samples
>
> **Reviewer Concern:**
> > It is unclear how much gain stems from the debate protocol itself versus just using more samples at training.
>
> We interpret this question in two ways:
>
> **If comparing against ground truth labels:** The comparison would be 3× samples with consensus methods vs 1× ground truth. However, since we work with self-generated data where quality varies, simply adding more uncurated self-generated samples could degrade rather than improve performance.
>
> **If comparing self-generation methods:** This is our focus—comparing single-round majority vote against debate-derived consensus, both using the same number of samples (3). Table 6 shows consistent improvements from debate-derived signals, demonstrating the value of the debate structure itself.
>
> Our ablations isolate debate's specific contributions:
> - Agent agreement increases from 27% to 43% through deliberation (Appendix P), indicating refined consensus rather than random aggregation
> - Debate trajectories contain evidence of error correction and peer grounding absent in independent samples
> - Figure 4 shows training on debate signals helps models learn effective peer utilization—base Llama-3B debate initially degrades performance on MathQA, but training corrects this failure mode
>
> These results validate that the deliberative process, not merely sample count, drives performance gains in both individual (Table 1) and multi-agent settings (Table 2).
>
> Finally, exploring additional aggregation methods (confidence-weighted voting, expert-weighted voting, LLM-as-judge) represents an interesting direction for future work beyond the scope of this paper.
>
> We truly appreciate your feedback in helping us improve the clarity of our presentation. If the clarifications we provided addressed your concerns, we kindly ask that you consider updating your score accordingly. Thank you!
>
> ---
>
> [1] ScPO - https://arxiv.org/pdf/2411.04109
> [2] TTRL - https://arxiv.org/pdf/2504.16084

---

### Official Review · Reviewer_Jz54 · 2025-11-05

**Soundness:** 1
**Presentation:** 3
**Contribution:** 2
**Rating:** 2
**Confidence:** 2

**Summary:**

This paper addresses Language Models’ reasoning inconsistencies stemming from probabilistic sampling by introducing a reinforcement learning framework MACA. MACA utilizes multi-agent debate to self-generate preference data, designating trajectories aligned with the majority consensus as 'preferred' and dissenting trajectories as 'not preferred'. It is then optimized on these self-supervised signals, using methods like DPO, to favor consensus-forming reasoning paths. It yields improvements in self-consistency and single-agent reasoning, and demonstrates generalization to unseen domains.

**Strengths:**

1.This paper tries to address an important problem: the reasoning inconsistency of LMs.

2.The MACA framework is conceptually simple.

3.The paper provides a very detailed appendix.

**Weaknesses:**

1.The core mechanism fails in face of “correct but minority” reasoning and actively rewards incorrect consensus.

2.The evidence for the central novelty claim is weak. As shown in Table 6, gains are marginal in 5 of 8 cases when compared fairly MV(C) with DMV(C).

3.The method essentially trains the model to agree more strongly with what it already agrees on, creating a self-reinforcing echo chamber that may amplify a model's inherent biases.

**Questions:**

See weaknesses.

---

> ### Author Response · Authors · 2025-11-19
> **Reply to Reviewer Jz54**
>
> We thank you for your feedback and appreciate that you found MACA conceptually simple with a detailed appendix. We address each concern below and hope you will reconsider your assessment given these clarifications.
>
> ## W1 & W3: "Correct but Minority" Reasoning and Echo Chamber Concerns
>
> **Reviewer Concerns:**
> > The core mechanism fails in face of 'correct but minority' reasoning and actively rewards incorrect consensus.
>
> > The method essentially trains the model to agree more strongly with what it already agrees on, creating a self-reinforcing echo chamber that may amplify a model's inherent biases.
>
> This theoretical concern about majority reasoning is inherent to consensus-based reinforcement methods—a recently established line of work including Debate-SFT [1], ScPO [2], TTRL [3]. While consensus signals may not always align with ground truth, empirical evidence demonstrates their effectiveness as self-generated training signals that correlate strongly with accuracy.
>
> We explicitly tested the echo chamber hypothesis, which predicts performance degradation as biases compound over iterations. Our iterative training analysis (Appendix H, Table 15) shows:
> - Iteration 2 consistently outperforms Iteration 1 across evaluation settings
> - Performance plateaus without collapse in Iterations 3 and 4
> - Iterations function as a tunable hyperparameter with diminishing returns
>
> If bias amplification were significantly impacting performance, we would observe degradation over iterations. Instead, we see improvement then stabilization. Furthermore, our generalization results (Table 3) show that training on math improves performance on unseen domains (+16.3% GPQA science, +11.6% CommonsenseQA), suggesting models learn transferable reasoning rather than domain-specific biases.
>
> ## W2: Evidence for Central Novelty Claim
>
> **Reviewer Concern:**
> > The evidence for the central novelty claim is weak. As shown in Table 6, gains are marginal in 5 of 8 cases when compared fairly MV(C) with DMV(C).
>
> We emphasize novelty in both single and multi-agent settings. For single agents, we use multi-agent debate signals as training data rather than single-round majority votes, outperforming prior majority-vote RL works like ScPO [2] and TTRL [3]. For multi-agent debates, while prior work uses SFT on debate traces to improve ensemble performance, we use RL (with extensive optimization studies) to achieve substantial gains over MV-SFT baseline, as shown in Tables 1-2. We'd like to highlight these multi-agent debate performance improvements, which may have been inadvertently overlooked in the initial review.
>
> ### Single-Agent Setting
>
> We acknowledge that our comparison to prior majority vote RL methods should have been more explicit. Updated Table 6 now directly compares MACA against:
> - **ScPO [2]:** MACA outperforms in all 8 settings tested
> - **TTRL [3]:** MACA outperforms in 6 of 8 settings, within SD in other 2 settings
>
> *Arrows show gains over instruction-tuned model. Statistics gathered over 3 seeds.*
>
> | Method | **MATH** |||| **GSM8K** ||||
> |--------|----------|----------|---------|----------|----------|----------|---------|----------|
> | | Qwen-2B | Llama-3B | Phi-4B | Llama-8B | Qwen-2B | Llama-3B | Phi-4B | Llama-8B |
> | TTRL | ↑**18.0**±2.9 | ↑5.3±5.7 | ↑6.1±2.1 | ↑7.5±0.2 | ↑**24.5**±1.6 | ↓22.3±8.3 | ↑4.8±1.6 | ↓13.5±5.7 |
> | ScPO | ↑2.3±1.1 | ↑3.4±0.2 | ↑0.1±0.5 | ↑3.7±0.6 | ↑15.9±3.4 | ↑1.6±0.8 | ↑5.9±0.5 | ↑12.9±1.2 |
> | MV-DPO | ↑16.7±0.4 | ↑**12.5**±0.7 | ↑**6.9**±0.2 | ↑**17.1**±0.8 | ↑23.8±0.9 | ↑**3.7**±0.3 | ↑**7.3**±0.3 | ↑**18.2**±0.2 |
>
> **Note:** ↑ indicates improvement over instruction-tuned model. Bold values indicate best performance within each column.
>
> These results show that using consensus from multi-agent debate with context is associated with improvements over single-round majority vote approaches.
>
> ### Multi-Agent Setting
>
> For multi-agent performance, we benchmark against prior work, which uses majority vote SFT [1]. Tables 1 and 2 compare MV-SFT with RL methods (GRPO, KTO, DPO) implemented within MACA:
> - MACA with DPO shows gains up to +42.7% (Llama3B-MathQA) and +26.9% (Llama8B-MATH)
> - Preference learning (DPO/KTO) consistently outperforms both GRPO and SFT when training on debate signals
>
> ### Figure 4 Analysis
>
> On Llama-3B MathQA, base model debate degrades performance versus initial-round majority vote. Post-training with MACA addresses this failure mode, with models showing improved ability to utilize peer feedback.
>
> Thank you for taking the time to review our paper. We believe that the clarifications and updates to the manuscript help better highlight the thoroughness and impact of our work. We would sincerely appreciate it if you would reconsider and increase your score accordingly.
>
> ---
>
> [1] Debate-SFT https://arxiv.org/abs/2501.05707
> [2] ScPO - https://arxiv.org/pdf/2411.04109
> [3] TTRL - https://arxiv.org/pdf/2504.16084

---

### Official Review · Reviewer_d4Qb · 2025-11-06

**Soundness:** 3
**Presentation:** 3
**Contribution:** 2
**Rating:** 4
**Confidence:** 4

**Summary:**

This paper proposes MACA (Multi-Agent Consensus Alignment), a post-training framework that “internalizes” self-consistency by using multi-agent debate to generate majority (consensus) and minority (dissent) trajectories, then optimizing the model with MV-DPO/MV-KTO/MV-GRPO or MV-SFT on those signals. The paper formalizes self-consistency via majority probability over sampled reasoning paths and measures agreement in multi-agent settings. Experiments on small LLMs (2B–8B) across math-heavy benchmarks report sizeable gains in sampling consistency and accuracy.

**Strengths:**

1. The paper explicitly defines single-agent sampling consistency and multi-agent agreement, making the target capability measurable, and uses them consistently in analysis.

2. MACA reuses standard post-training objectives (DPO/KTO/GRPO/SFT) with debate-derived preference signals, and shows that MV-DPO/KTO tend to outperform SFT and scalar-reward baselines across several model/dataset pairs.

3. The paper compares debate-majority supervision to ground-truth labels and finds similar performance, and also tests training with/without peer context during debate—both helpful for understanding what drives gains.

**Weaknesses:**

1. Since self-consistency prompting (Wang et al., 2022) is a main comparator, I expected a training-time baseline that (i) uses self-consistency/majority vote to curate a dataset (e.g., majority-consistent rollouts), then (ii) finetunes/SFTs on that curated set—without multi-agent debate. This would test whether debate-derived signals truly add value over classical self-consistency data augmentation.

2. The training/inference cost versus gains isn’t quantified (GPU hours, wall-clock, debate throughput).

3. Most training is math-centric; generalization to GPQA/CSQA is interesting but still limited in breadth. Also, the “formalization of self-consistency” largely re-casts majority probability/consensus ideas already known from self-consistency and majority-vote literature; the novelty is primarily in using debate-derived preference pairs, which would be stronger if contrasted directly against the SC-curation+FT baseline noted above.

**Questions:**

As in weaknesses

---

> ### Author Response · Authors · 2025-11-19
> **Reply to Reviewer d4Qb**
>
> Thank you for your thoughtful review. We appreciate your recognition of our explicit consistency metrics, finding that preference learning outperforms GRPO for majority vote reinforcement, and thorough ablations on debate dynamics. We address your concerns below.
>
> ## W1: SC-curation+FT Baseline Comparison
>
> **Reviewer Request:**
> > I expected a training-time baseline that (i) uses self-consistency/majority vote to curate a dataset, then (ii) finetunes/SFTs on that curated set—without multi-agent debate.
>
> **Reviewer Concern:**
> > The novelty is primarily in using debate-derived preference pairs, which would be stronger if contrasted directly against the SC-curation+FT baseline.
>
> We have included this exact comparison in our paper. We acknowledge that our comparison to prior majority vote methods should have been more explicit, and thank the reviewer for bringing this up. Updated Table 6 now directly compares MACA against:
>
> - **ScPO** (single-round majority vote DPO without context): MACA outperforms in all 8 settings tested
> - **TTRL** (single-round majority vote GRPO): MACA outperforms in 6 of 8 settings, within SD in other 2 settings
>
> *Arrows show gains over instruction-tuned model. Statistics gathered over 3 seeds.*
>
> | Method | **MATH** |||| **GSM8K** ||||
> |--------|----------|----------|---------|----------|----------|----------|---------|----------|
> | | Qwen-2B | Llama-3B | Phi-4B | Llama-8B | Qwen-2B | Llama-3B | Phi-4B | Llama-8B |
> | TTRL | ↑**18.0**±2.9 | ↑5.3±5.7 | ↑6.1±2.1 | ↑7.5±0.2 | ↑**24.5**±1.6 | ↓22.3±8.3 | ↑4.8±1.6 | ↓13.5±5.7 |
> | ScPO | ↑2.3±1.1 | ↑3.4±0.2 | ↑0.1±0.5 | ↑3.7±0.6 | ↑15.9±3.4 | ↑1.6±0.8 | ↑5.9±0.5 | ↑12.9±1.2 |
> | MV-DPO | ↑16.7±0.4 | ↑**12.5**±0.7 | ↑**6.9**±0.2 | ↑**17.1**±0.8 | ↑23.8±0.9 | ↑**3.7**±0.3 | ↑**7.3**±0.3 | ↑**18.2**±0.2 |
>
> **Note:** ↑ indicates improvement. Bold values indicate best performance within each column.
>
> Tables 13-14 (Appendix F) provide comprehensive comparisons of the impact of including debate context during RL across all post-training methods. For comparisons against SFT, Tables 1 and 2 show that MV-SFT consistently underperforms our RL variants, so we felt single-round majority vote RL methods comparison was more relevant. We would also like to note, as highlighted in our general response to all the reviewers, that aside from the single-agent setting, a key area of novelty is the use of RL to improve multi-agent debate performance itself, where prior works primarily use SFT to reinforce majority traces [1] [2].
>
> ## W2: Training/Inference Cost Quantification
>
> **Reviewer Concern:**
> > The training/inference cost versus gains isn't quantified (GPU hours, wall-clock, debate throughput).
>
> **Training Cost:** We added cost-performance analysis to Appendix Figure 13 (p.40). In our single-agent experiments, MV-DPO with debate achieves the best efficiency-performance tradeoff: comparable or superior gains to TTRL (which requires 2.2-7.7 GPU hours) using only 0.73-1.68 GPU hours, while outperforming ScPO (0.075-0.58 GPU hours) in accuracy gains.
>
> **Inference Cost:** Post-training internalizes self-consistency benefits, eliminating inference overhead. Both single-agent (Figure 3) and multi-agent debate (Figure 4) achieve performance gains with no additional inference time compared to standard pre-training variants.
>
> ## W3: Generalization Breadth and Novelty
>
> **Reviewer Concern:**
> > Most training is math-centric; generalization to GPQA/CSQA is interesting but still limited in breadth.
>
> We use representative reasoning datasets standard in this area of research, and as highlighted in Table 3, our results show consistent cross-domain transfer to non-math datasets after training on any of the math datasets (Math→Science: +16.3pp GPQA; Math→Commonsense: +11.6pp CSQA; Math→Unseen Math: +27.7pp). We provide further experiments in Tables 17 and 18 for generalization with debate compared against ground truth within the MACA framework.
>
> Thank you for all your feedback! We believe that the clarifications and adjustments we made address the main concerns raised in your review. We kindly ask that you reconsider and update your score accordingly, and please feel free to let us know if any additional clarification would be helpful.
>
> ---
>
> [1] https://arxiv.org/pdf/2402.01620
> [2] https://arxiv.org/abs/2501.05707

---

### Official Review · Reviewer_eWWi · 2025-11-06

**Soundness:** 3
**Presentation:** 3
**Contribution:** 3
**Rating:** 8
**Confidence:** 3

**Summary:**

The paper introduces MACA (multi-agent consensus alignment), a reinforcement learning framework that trains LLMs to be more self-consistent. MACA uses debate-derived consensus trajectories and trains LLMs via preference learning (DPO/KTO) or other objectives (GRPO, SFT).

Experiments show significant performance gains on math reasoning datasets and strong generalization to unseen reasoning domains, validating that MACA can improve self-alignment and elevate reasoning capabilities.

**Strengths:**

- The work is well-motivated, addressing LLMs' inconsistency when sampled multiple times. The proposed method is simple yet effective, using majority voting from multi-agent debates as a weak supervision signal to construct preference pairs for training.
- Thorough experiments and ablations yield valuable insights, such as "Multi-agent debate produces more informative training signals than single-round majority voting" and "Addressing consensus alignment through preference learning improves over GRPO and SFT".

**Weaknesses:**

No major weaknesses. A few questions are listed below.

**Questions:**

- Regarding the scaling of debate settings, will **more agents** or **more rounds** or **heterogeneous agents** yield better consensus? Would scaling these dimensions lead to higher-quality pairwise training data?

- Cross-model transfer: If debate trajectories generated by a more capable LLM (e.g., Llama-8B) are used to train a smaller model (e.g., Llama-3B), how would this impact the smaller model’s **self-consistency** and accuracy?

- Presentation: In Figure 2, which post-training methods (DPO, KTO, or SFT) are being illustrated? In Table 4, what does “Debate” refer to in the single-agent setting?

---

> ### Author Response · Authors · 2025-11-21
> **Reply to Reviewer eWWi**
>
> We thank the reviewer for the positive assessment and for recognizing MACA's motivation, effectiveness, and thorough experimental validation. We appreciate your insightful questions and address each below.
>
> ## Q1: Scaling Debate Settings
>
> **Reviewer Question:**
> > Will more agents or more rounds or heterogeneous agents yield better consensus? Would scaling these dimensions lead to higher-quality pairwise training data?
>
> **More agents/trajectories:** Figure 3 demonstrates that sampling more trajectories improves majority vote performance (with diminishing returns). Since deploying more agents would naturally generate more trajectories, we expect this would similarly improve consensus signal quality.
>
> **More rounds:** Prior work [1] has shown that additional deliberation rounds improve debate performance. Our results are consistent with this—we observe that the additional debate round increases inter-agent agreement rates (Figure 12, Appendix P), suggesting improved consensus quality.
>
> **Heterogeneous agents:** Our framework theoretically supports heterogeneous agents, though we maintained homogeneous agents for controlled self-improvement analysis. We hypothesize that diverse model capabilities could inject beneficial diversity into debates, potentially improving consensus robustness. However, developing rigorous methods to measure such effects remains an interesting direction for future work.
>
> ## Q2: Cross-Model Transfer
>
> **Reviewer Question:**
> > If debate trajectories generated by a more capable LLM (e.g., Llama-8B) are used to train a smaller model (e.g., Llama-3B), how would this impact the smaller model's self-consistency and accuracy?
>
> We conducted experiments to explore MACA's potential for knowledge distillation, training Llama-3B on debate outputs from Llama-8B:
>
> ### Llama-3B Results
>
> | Training Signal Source | Model | MATH | GSM8K | Method |
> |---|---|---|---|---|
> | None | Llama-3B | 32.0 | 69.6 | Instruction-tuning |
> | Llama-3B (self) | Llama-3B | 51.4 | 76.4 | MV-DPO |
> | Llama-8B (teacher) | Llama-3B | 41.0 | 71.4 | MV-SFT |
> | Llama-8B (teacher) | Llama-3B | 47.0 | 77.8 | MV-DPO |
>
> Results demonstrate MACA's effectiveness as an RL-based distillation technique: (1) MV-DPO significantly outperforms standard SFT-based distillation (47.0 vs 41.0 on MATH), confirming that debate-derived preference learning provides superior supervision over SFT approaches, and (2) self-generated signals achieve competitive performance (51.4 on MATH) without requiring a teacher model. While the relative benefits of self versus teacher debate signals merit further investigation, we focus on self-improvement without external supervision as our primary contribution.
>
> ## Q3: Presentation Clarifications
>
> **Reviewer Question:**
> > In Figure 2, which post-training methods (DPO, KTO, or SFT) are being illustrated? In Table 4, what does 'Debate' refer to in the single-agent setting?
>
> We've updated the manuscript for clarity:
> - **Figure 2:** MV-DPO consistency curves are illustrated - this was the most performant method in general.
> - **Table 4 "Debate":** Refers to multi-agent debate. For complete comparisons:
>   - Single-agent (GT vs. DMV): Tables 11, 17, 18
>   - Multi-agent (GT vs. DMV): Table 12
>
> Thank you for your constructive feedback that helped strengthen our work.
>
> ---
>
> [1] (Du et al., 2023)

---

> > ### Comment · Reviewer_eWWi · 2025-11-27
> >
> > Thank you for the response. I have updated my review accordingly.

---

### Author Response · Authors · 2025-11-19
**General Response to All Reviewers and AC**

We sincerely thank all reviewers and the AC for your thoughtful engagement with our work. Your feedback has been invaluable in clarifying our contributions and addressing presentation concerns.

## Core contributions: RL Post-Training for Multi-Agent Debate

MACA is a framework for improving both the consistency and performance of language model reasoning. The key contribution is using consensus signals from multi-agent debate—rather than single-round majority vote—as gold labels for RL, while including debate context that reveals peer agents' chains-of-thought and evidence of relative grounding. Through this process, MACA achieves three outcomes: (1) improved consistency of individual agents, as measured by sampling consistency metrics, (2) improved performance of individual agents within the debate, and (3) enhanced performance of the multi-agent debate system overall.

## Performance Impact in Single-Agent Setting & Comparison to other works

For the single-agent setting, recent work has shown the value of reinforcing majority vote with RL through ScPO [2] (using DPO) and TTRL [3] (using GRPO). While our original submission did include comparisons between MACA and these methods (with and without debate context) in principle throughout the ablation section and appendix, we agree with the reviewers that this comparison should have been made more explicit and easily accessible to the reader.

We thank the reviewers for providing this valuable feedback, and have accordingly updated Table 6 to directly compare MACA's MV-DPO (RL on debate majority vote with debate context) against ScPO (single-round majority vote with DPO) and TTRL (single-round majority vote with GRPO). Across 3 seeds, a single agent trained by MACA outperforms ScPO in all 8 settings tested and TTRL in 6 of 8 settings (within SD in the other 2). This demonstrates clear benefits of MACA over single-round majority vote, with correspondingly larger performance improvements over the base instruction-tuned model.

To further clarify how MACA generalizes these approaches within its framework, we have: (1) added a line in Related Work explaining how these alternative majority vote RL methods can be generalized as special cases of the MACA framework, (2) included an additional paragraph in Methods detailing this relationship, and (3) updated the relevant paragraph for Table 6 in the discussion section. These additions make explicit that MACA provides a more general framework that encompasses and extends beyond existing single-round majority vote approaches.

## Performance Impact in Multi-Agent Setting

For the multi-agent setting, we benchmark MACA against prior work that uses SFT to reinforce consensus-reaching debate traces [1]. Tables 1 and 2 compare MV-SFT with three RL methods (GRPO, KTO, DPO) implemented within MACA. As Reviewer @d4Qb noted, preference learning formulations (DPO/KTO) consistently outperform both scalar reward RL (GRPO) and SFT, yielding substantial improvements over prior multi-agent debate methods.

To understand how debate context enhances performance, we conduct extensive ablations. Tables 13-14 compare post-training methods with and without debate context across models and datasets, with Table 5 providing a focused view for MV-DPO. Figure 4 demonstrates three key findings: (1) post-training improves performance at every debate stage, (2) models learn to better utilize peer feedback, as shown by improved final-round performance when given previous-round responses, and (3) while debate can degrade small model performance, RL mitigates this issue.

Finally, iterative training with MACA continues to improve performance beyond the first iteration (Table 15), though with diminishing returns but no notable performance collapse.

## Summary

MACA advances multi-agent debate from an inference-time technique to an effective training framework. We provide comprehensive comparisons demonstrating: (1) RL's superiority over SFT-based methods used in prior multi-agent work, (2) explicit benchmarking against single-agent majority vote methods (ScPO/TTRL), and (3) preference learning as the most effective approach for consensus alignment.

Our key insight is that training on debate signal with the debate context provides a stronger training signal than majority vote of independent samples. This improves both reasoning performance and the models' ability to leverage debate dynamics. The updated manuscript makes these comparisons explicit while maintaining the thoroughness of our original analysis.

---

[1] Debate-SFT https://arxiv.org/html/2501.05707v1
[2] ScPO - https://arxiv.org/pdf/2411.04109
[3] TTRL - https://arxiv.org/pdf/2504.16084

---

### Note · Program_Chairs · 2026-01-17
**Submission Desk Rejected by Program Chairs**

The following references in this submission do not refer to real documents and/or have major errors in bibliographic information:

 C. J. Nemeth. The Influence of Minority Social Position on the Formation of Social Judgment. Psychology Press, 1985.